# Probing the chemical stability between current collectors and argyrodite Li$_6$PS$_5$Cl sulfide electrolyte
Artur Tron [1] ✉, Alexander Beutl[1], Irshad Mohammad[1] & Andrea Paolella[1,2] ✉

Recently, sulfide-based electrolytes, including the argyrodite family (Li$_6$PS$_5$X, X = Cl, Br, I), are considered promising candidates for all-solid-state battery fabrication due to their high ionic conductivity. However, from the industrial point of view, other parameters such as the chemical and electrochemical stability toward current collectors are equally important, but often neglected. Although many efforts have been directed toward the investigation, optimization and testing of sulfide electrolytes into a press device (10 MPa) with a stainless-steel current collector, the investigation of the current collector's behavior in contact with sulfide solid electrolytes in coin cell (0.2 MPa) or pouch cell (0.1-0.2 MPa) formats is still an open question. In this work, the systematic physicochemical and electrochemical analyses of copper, nickel, stainless steel, aluminum, and aluminum-carbon current collectors in contact with the Li$_6$PS$_5$Cl electrolyte in coin cell format configuration is reported, enabling the understanding of the reaction mechanisms. While SS, Ni, Al and Al/C show good chemical stability, Cu, Li, and Cu/Li have high corrosion susceptibility in sulfide electrolytes. Therefore, this study supports the selection of appropriate current collectors for fabricating sulfide-based components, especially via the wet chemistry process which is a promising approach for the industrialization of solid-state batteries with sulfide electrolyte.

Conventional lithium-ion batteries (LIBs) are extremely important in our daily lives due to the very high volume and energy densities of these types of batteries compared to other rechargeable batteries, which have led to an enormous increase in the use of these batteries in various applications[1–3]. Solid-state batteries (SSBs) with ceramic solid electrolytes (e.g. sulfides and halides) have received a great deal of attention in the last half-decade[4,5]. However, in literature, the In/Li alloy has been used as anode in these systems due to its high stability[6,7]. The reactivity of sulfide electrolytes toward lithium metal and current collectors is still an open question due to the toxic and possible reaction of sulfide components with pure alkaline and transition metals, which can lead to corrosion and result in general capacity degradation of the cell. The manufacture of conventional lithium-ion and solid-state batteries consists mainly of active electrode materials, separators for LIBs, solid electrolytes as separators and electrolytes for SSBs, and current collectors[8–12]. It should be noted that numerous studies have been dedicated to optimising and improving active electrode materials and electrolytes in recent decades. Unfortunately, only a few studies have been carried out on current collectors for solid-state batteries with sulfide

electrolytes. In the field of lithium-ion batteries, aluminium (Al) foils current are typically used as current collectors at the positive electrodes while copper (Cu) foils are used as current collectors at the negative electrodes. About corrosion in LiBs most studies have focused on understanding the corrosion and degradation mechanisms of current collectors for high-voltage cathodes and ionic liquid systems[13–22]. In/Li anode is the most explored configuration in solid-state batteries in various press and coin cell formats in combination with sulfide ceramic electrolytes[23–27]. Considering the nature of the solid sulfide electrolytes, especially from a chemical and electrochemical point of view, using the positive electrode fabrication and contact with lithium metal anode—the current collectors should possess high electrical conductivity to reduce the cell resistance and electrochemical and chemical stability in contact with sulfide electrolytes and cathode composite with sulfides, and lithium metal anode in the operating potential window of NCM electrodes[28–30]. Previous works[31,32] have shown that solid sulfide electrolytes can react with Cu current collectors to produce undesirable side reaction components between copper (Cu) and the sulfide electrolyte Li$_6$PS$_5$Cl can lead to the formation of copper sulfides (Cu$_x$S/Cu$_2$S and/or CuS) and

[1]AIT Austrian Institute of Technology GmbH, Center for Transport Technologies, Battery Technologies, Giefinggasse 2, 1210 Vienna, Austria. [2]Dipartimento di Scienze Chimiche e Geologiche Università degli Studi di Modena e Reggio Emilia Via Campi 103, Modena, 41125, Italy. ✉e-mail: artur.tron@ait.ac.at; andrea.paolella@unimore.it

phosphides ($Cu_3P$). These compounds degrade the electrolyte's ionic conductivity and compromise the electrochemical stability of the system. It should be noted that the reaction mechanism and the formation of side reaction components using sulfide electrolytes in contact with different types of current collectors and a lithium metal anode are still unclear and have not been fully investigated. In this work, different current collectors (Cu, SS, Ni, Al and Al/C) and lithium metal sources (Li and Cu/Li) were compared in order to understand their chemical and electrochemical effects with solid-state batteries with $Li_6PS_5Cl$ sulfide electrolytes in coin cell configuration. The formation of side reactions that could potentially disrupt the balance of electronic and ionic conduction within the battery leading to capacity fading is discussed from the point of view of chemical and electrochemical stability of current collectors.

## Results and discussion

Prior to the screening of the current collectors in contact with $Li_6PS_5Cl$ sulfide electrolyte in solid-state battery cells in half (CC|LPSCl|CC) and full (CC|CAM|LPSCl|Li|CC) configurations, it was used the commercial solid sulfide electrolytes used for all-solid-state batteries[26,27], this material was characterized by the physicochemical and electrochemical methods as shown in Fig. S1 (Supplementary Information). It's important to underline that the commercial $Li_6PS_5Cl$ sulfide electrolyte didn't show any variations and its XRD pattern matches well with that of the lithium argyrodite $Li_7PS_6$ system (JCPDS No. 34-0688) and belongs to the F-43m space group[28]. The morphologies of the sulfide solid electrolyte sample, which has a rough surface with a size of about 10-15 μm, are shown in Fig. S1b (Supplementary Information). To confirm the presence of P, S and Cl elements, the EDS method was carried out. It shows that the molar ratio is close to 1:5:1 of argyrodite structure. The ionic conductivity of this solid electrolyte has been measured in pellet form, as illustrated in Figs. S2 and S3 (Supplementary Information). Additionally, it has been evaluated when sandwiched between Li-ion blocking stainless steel (SS) disks in coin cell format CR2032 (Table S1, Supplementary Information), as shown in Fig. S4 (Supplementary Information). The obtained Nyquist plots don't show any additional semicircles, which could be related to the grain boundary resistances associated with the reduced particle-particle contact area (Fig. S1c, Supplementary Information) This results in higher impedance values and lower ionic conductivities compared to the untreated (pristine powder does not undergo solvent treatment) sample and is observed for solid electrolytes after solvent treatment[33,34]. The ionic conductivity of $Li_6PS_5Cl$ powders shows a similar trend to the sulfide electrolytes under pressure as reported in previous works[23,35]. In addition, the density of the obtained powder pellet (densified at 300 MPa), used for the measurement of ionic conductivity, is compared and a significantly lower density (1.49 g cm$^{-3}$) is obtained for the theoretical value of 1.64 g cm$^{-3}$ [36]. This can be explained by differences in the mechanical properties of the obtained material from the theoretical value in terms of hardness and plasticity during densification by cold pressing, which can result in significant differences in the quality of the prepared powder pellets, and consequently, samples with contaminants appear to have a higher degree of defects and porosity. This could be a further explanation for the slightly lower ionic conductivity and the type of method to be used for the measurement of ionic conductivity and density values[26,27,33,35].

In order to determine the chemical reactivity of current collectors (Cu, SS, Ni, Al and Al/C), LPSCl sulfide electrolyte and current collectors have been in contact for 24 h in coin cell format (CR2032) as shown in Fig. 1 and Fig. S4 (Supplementary Information). It was found that after 24 h of direct contact, a degradation of Cu current collector is observed while the other SS, Ni, Al and Al/C collectors don't have any visible surface changes (Fig. 1a-e). As shown in Fig. 1f, g, after 2 months of direct contact with $Li_6PS_5Cl$ and Cu current collector observed the high chemical reactivity and high surface changes from both sides as front and back with possible formation of side reaction components (CuS, $Cu_2S$, $Cu_3P$ / CuP, $Li_3P$, $Li_2S$, and LiCl) which can be associated to the chemical corrosion[30,32,37–39]. After 24 h of contact with Cu foil and sulfide electrolyte, Cu foil showed the presence of reddish areas, after 2 months front and back sides were covered the corrosion

products with red colour gradually transformed into bluish-purple, yellow and grey-black from direct contact with sulfide electrolyte. A similar effect was observed for Cu foil at heat treatment in an air atmosphere at a temperature of 100 °C[37,40]. Even, the non-direct contacted area of Cu foil undergoes the chemical reactions from sulfide electrolyte with comprehensive surface and structure changes. Subsequently XRF analysis were performed on these samples 2 months aged, where it was found that the P, S, Cl and Cu have high values compared to the pristine Cu current collectors before contact with sulfide electrolyte which corresponds to the formation of side reaction components (Fig. 1g). On the contrary for Al current collector low values of P, S, Cl and Al were observed suggesting that aluminium current collector could exhibit high chemical stability toward sulfides-based electrode as shown in Figure S5 (Supplementary Information).

To further understand and confirm possible chemical reactions which form on the surface current collectors were carried out the structure analysis via the XRD analysis as shown in Fig. 2. As shown in Fig. 2a, it was found that the Cu foil exhibited an obvious reaction with the sulfide SE after aging with possible side reaction formation such as $Cu_2S$ ($Cu_xS$ of Cu–S compounds, PDF #98-901-6669), and its colour remained changed (Fig. 1a, f). In addition, the main peaks of Cu (PDF #98-005-2256) shifted at lower 2θ angles due to the chemical reactions and form side products due to the corrosion of the Cu current collector, indicating a slow reaction between the copper current collector and the sulfide solid electrolyte[32,37]. While this effect becomes more pronounced in the XRD pattern and on the surface of the copper current collector after one week. As reported in the literature, it was confirmed the formation of $Cu_2S$ compounds on Cu foil surfaces under different corrosive conditions with surface changes in colour can have a high effect on the electrochemical performance of solid-state battery cell with cathode active material[32]. While for SS and Ni current collectors, the peaks corresponding to the iron (PDF #98-063-1730) and Ni (PDF #96-151-2527) showed no impurities or peak shift: these results confirm the high stability of SS and Ni foil to the chemical reactivity of $Li_6PS_5Cl$ sulfide electrolyte (Fig. 2b, c). Regarding the Al and Al/C current collectors, it should be noted that no observed any visible surface changes from Fig. 1d, e, however, XRD analysis shows small structure changes with chemical reactions forming on the surface of Al (PDF #98-015-0692) and Al/C (PDF #98-005-3781) foils with small shifted main peaks (Fig. 2d, e) which can be associated with possible formation side reaction components of $Al_2S_3$ or $CS_2$ or similar (Al–S and C–S compounds)[41]. Obtained results of current collectors in direct contact with sulfide electrolytes suggest minor or moderate corrosion of these collectors. Subsequently, lithium metal (Li with a thickness of around 100 μm) and Cu/Li metal (bi-layer, Cu of 15 μm and Li of 40 μm) anodes were tested with sulfide electrolytes[26,27]. Figure 2f, g shows the XRD pattern of the pristine Li foil and the Cu/Li metal anode, respectively: Three crystalline peaks (PDF #98-064-2104) can be indexed to Li metal for both samples before contacting with LPSCl electrolyte. While, after contact with LPSCl solid electrolyte in 24 h, it was found the extra peaks for pristine Li foil related to the formation of some impurity phases of $Li_2O$ (PDF #98-002-6892)/H-Li-O (PDF #98-010-8886)[42–44] that can lead to chemical corrosion and these results related to the chemical stability of samples (Fig. 3). Moreover, the formation of $Li_2O$ on the surface of the lithium metal anode after LPSCl contact may be attributed to the native SEI, which is conventionally described as having a multiphasic structure with fully reduced, dense ionic phases (such as $Li_2O$ and/or LiF) closest to the Li interface in the "inner layer." This SEI exhibits nanoscale thickness, similar to the effects observed in lithium-ion battery systems with liquid organic electrolytes[42]. In addition, the Cu/Li sample (Cu foil, PDF #98-062-7113) shows the same images, but more aggressive corrosion with the formation of side reaction components such as Cu-P (PDF #98-010-8396) and shifted main peaks of Li and Cu are observed. Thus, the main peaks corresponding to the chemical reactivity of lithium metal and Cu/Li anodes to the LPSCl electrolyte are newly grown/shifted during time by direct contact, which can confirm the susceptibility of pure Li metal to the $H_2S$ gas generated by the LPSCl electrolyte decomposition[4,30,32]. Based on the obtained XRD results, the possible formation of chemical reactions between the $Li_6PS_5Cl$ sulfide solid

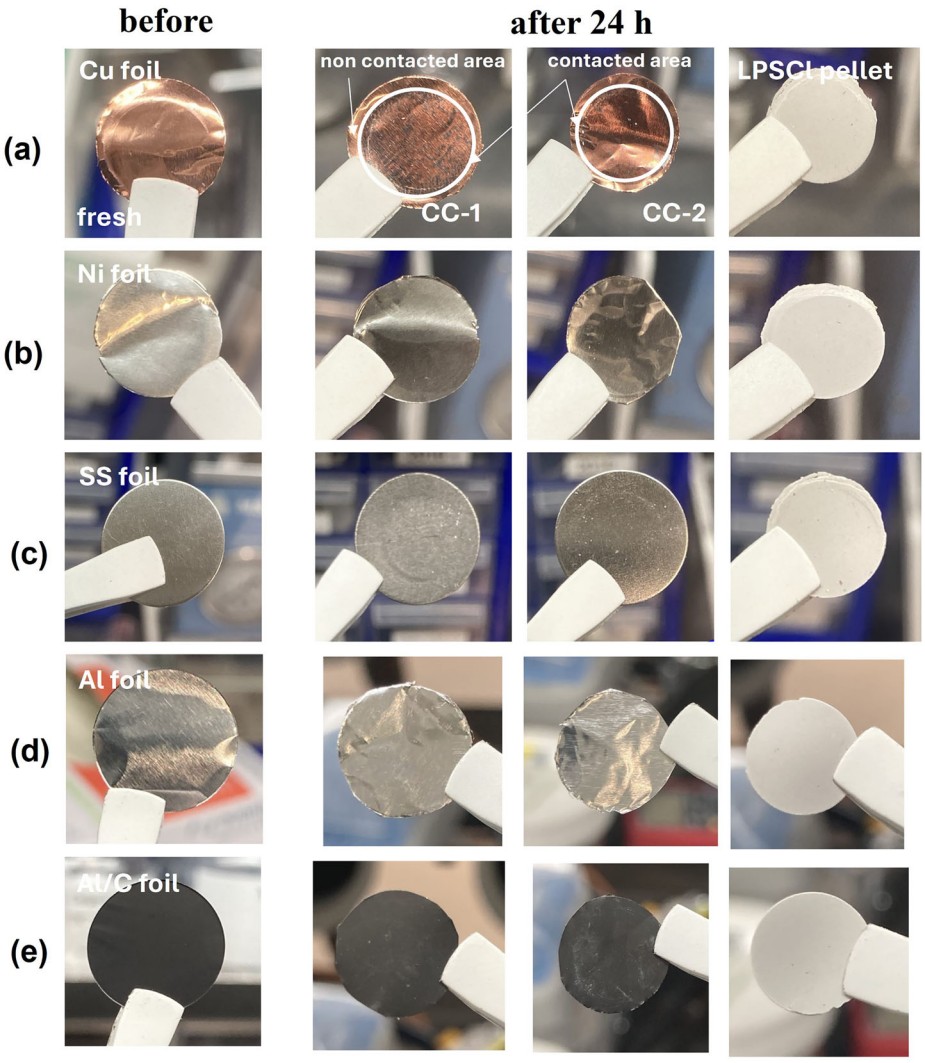

## after 2 months contacted with LPSCl

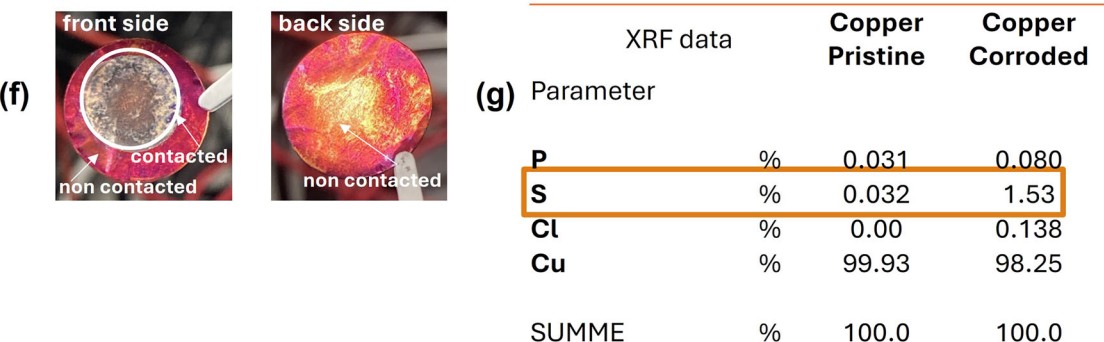

| XRF data Parameter | | Copper Pristine | Copper Corroded |
|---|---|---|---|
| **P** | % | 0.031 | 0.080 |
| **S** | % | 0.032 | 1.53 |
| **Cl** | % | 0.00 | 0.138 |
| **Cu** | % | 99.93 | 98.25 |
| SUMME | % | 100.0 | 100.0 |

**Fig. 1 | Photos of current collectors before and after contact with LPSCl.** Photos of current collectors of **a** Cu, **b** SS, **c** Ni, **d** Al, and **e** Al/C before and after contact (in direct contact into coin cell CR2032 in symmetric system of CC-1|LPSCl|CC-2) with LPSCl electrolyte for 24 h and after 2 months: photos of Cu current collector of **f** after contact with LPSCl electrolyte for 24 h, and **g** XRF analysis after 2 months of front electrolyte and various current collectors (Cu, SS, Ni, Al, Al/C, Cu/Li, and Li metal anode) can be considered. The following chemical reactions and their byproducts are proposed (1–6):

and back side of Cu current collector. This experiment simulates direct contact (into coin cell CR2032) of LPSCl pellet and current collectors and the possible formation of side reaction components (impurity phases of $Li_2S$, LiCl, and/or $Li_3PO_4$) into coin cell CR2032 which contacted with current collectors and lithium metal anode sources.

$$Cu : Li_6PS_5Cl + Cu \rightarrow Li_3P + Li_2S + Cu_2S + LiCl; \quad (1)$$

$$SS : Li_6PS_5Cl + Fe \rightarrow Li_3P + Li_2S + FeS + LiCl; \quad (2)$$

$$Ni : Li_6PS_5Cl + Ni \rightarrow Li_3P + Li_2S + NiS + LiCl; \quad (3)$$

$$Al : Li_6PS_5Cl + Al \rightarrow Li_3P + Li_2S + Al_2S_3 + LiCl; \quad (4)$$

**Fig. 2 | XRD patterns of current collectors before and after contact with LPSCl.** XRD patterns of current collectors of **a** Cu, **b** SS, **c** Ni, **d** Al, **e** Al/C, **f** Li and **g** Cu/Li before and after direct contact with LPSCl electrolyte for 24 h.

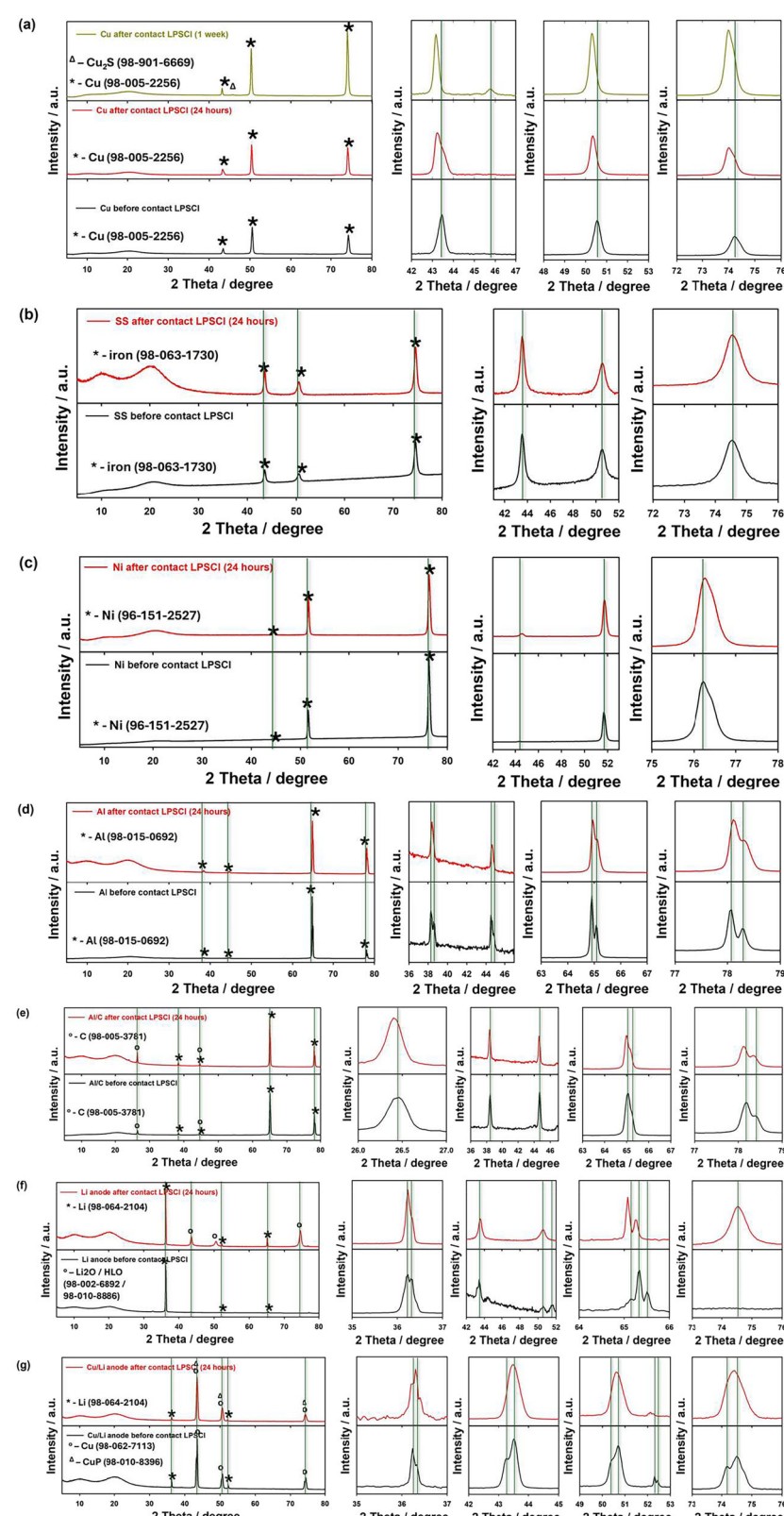

$$Al/C : Li_6PS_5Cl + Al/C \rightarrow Li_3P + Li_2S + Al_2S_3 + LiCl + C; \quad (5)$$

$$Li \text{ and/or bi} - \text{layer } Cu/Li : Li_6PS_5Cl + Li \rightarrow Li_3P + Li_2S + LiCl. \quad (6)$$

These reactions arise from the decomposition of the $Li_6PS_5Cl$ sulfide solid electrolyte, resulting in the formation of various sulfides, phosphides,

and chlorides[30,45–47]. Therefore, the specific byproducts depend on the reactivity of the current collector material with the solid electrolyte. However, further detailed investigation is required to confirm these chemical reactions.

To explore the evolution of surface chemistry for different cycled current collectors, XPS spectra were acquired from the electrolyte-facing side of the current collectors after cycling. Figure 3 displays the XPS spectra

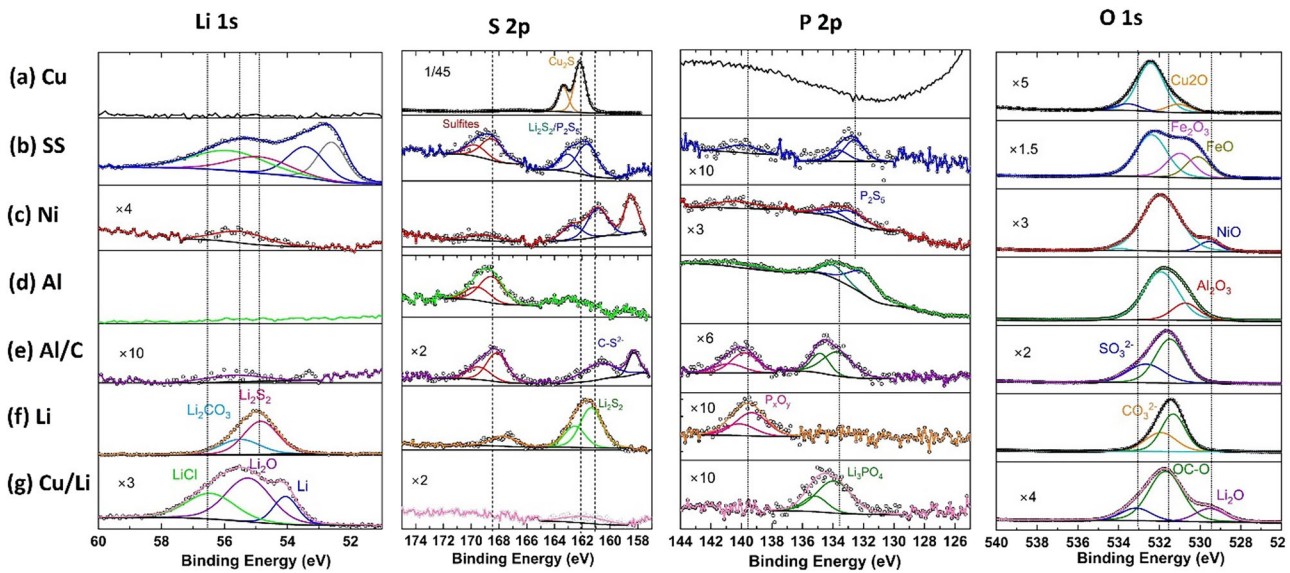

**Fig. 3 | XPS spectra of current collectors.** Deconvoluted XPS detail spectra for the Li 1 s, S 2p, P 2p, and O 1 s signals of (**a**) Cu, (**b**) SS, (**c**) Ni, (**d**) Al, (**e**) Al/C, (**f**) Li, and (**g**) Cu/Li current collectors.

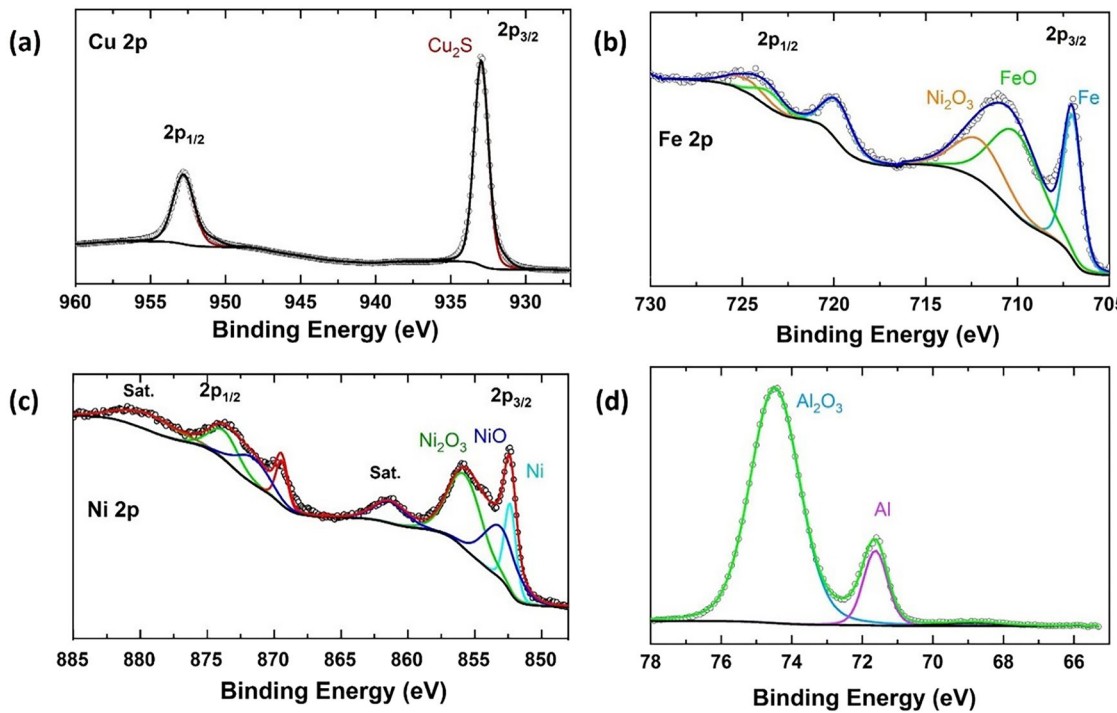

**Fig. 4 | high resolution XPS spectra.** The high-resolution deconvoluted XPS spectra of **a** Cu 2p for Cu, **b** Fe 2p for SS, **c** Ni 2p for Ni, and **d** Al 2p for Al current collectors.

of Cu, SS, Ni, Al, Al/C, Li, and Cu/Li, focusing on the Li 1s, S 2p, P 2p, and O 1s regions. The core XPS spectrum of the Cu current collector is shown in Fig. 3a. As expected, no component of Li was detected in the Li 1s spectrum for Cu. The high-resolution S 2p spectrum for Cu reveals a significant doublet located at 161.7 eV (S $2p_{3/2}$) and 162.9 eV (S $2p_{1/2}$), attributed to the $Cu_2S$ compound[48]. Confirmation of the presence of $Cu_2S$ is also observed in the Cu 2p spectrum (Fig. 4a)[49]. This finding aligns with the results from the XRD pattern (Fig. 2a). The formation of $Cu_2S$ typically occurs when the decomposed species $S^{2-}$ from the $Li_6PS_5Cl$ electrolyte reacts directly with Cu. Its formation is usually undesirable as it degrades the interface, leading to increased resistance and mechanical issues (volume changes) at the interface. As seen in the Li 1s spectrum of the SS current collector (Fig. 3b), several peaks related to lithium components were observed. One peak

located at 56.0 eV corresponds to the LiCl compound[50]. A doublet at 198.5 eV ($2p_{3/2}$) and 200.0 eV ($2p_{1/2}$) is observed in the Cl 2p region, confirming the presence of LiCl (Fig. S6a, Supplementary Information). Another peak in the Li 1 s spectrum at 55 eV for SS is likely related to $Li_2S_2$[51]. Additionally, traces of sulfite and $P_2S_5$ compounds are detected at 168.8 eV and 161.6 eV ($2p_{3/2}$) in the S 2p region, respectively. The XPS spectra of the Ni current collector show a trace of $P_2S_5$ in both the S 2p and P 2p regions at binding energies of 132.9 eV and 160.9 eV ($2p_{3/2}$)[52]. The Al and Al/C current collectors exhibit similar species, including sulfite and $PO_4^{3-}$ in the S 2p and P 2p spectra, as depicted in Fig. 3d, e. For the Li metal current collector, $Li_2CO_3$ was identified in both the Li 1 s and C 1 s spectra (Fig. S6b, Supplementary Information) at binding energies of 55.6 eV and 290.2 eV, respectively[53]. The formation of $Li_2CO_3$ on the surface of the lithium anode

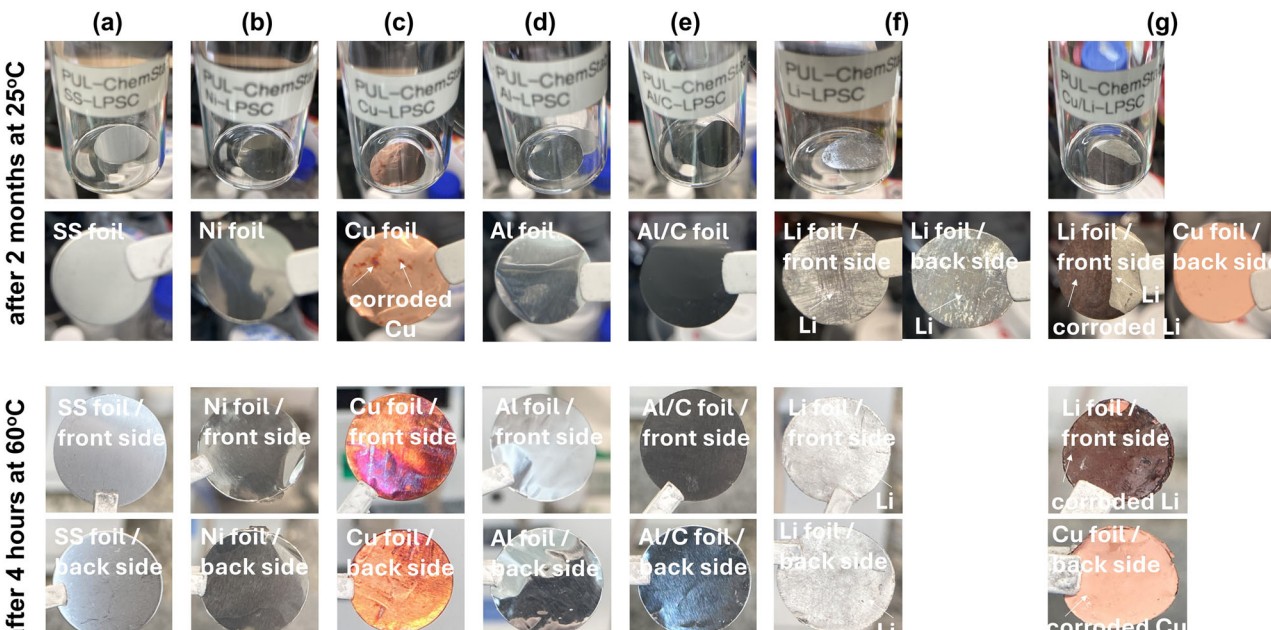

**Fig. 5 | Photos of current collectors after extended treatment with LPSCl and after treatment and elevated temperatures.** Photos of **a** SS, **b** Ni, **c** Cu, **d** Al, **e** Al/C, **f** Li and **g** Cu/Li current collectors after contact (undirect contact into vials) with LPSCl electrolyte after 2 months and after 4 h heat treatment at 60 °C. This experiment simulates the contact of LPSCl pellet and current collectors and the possible formation of side reaction components (impurity phases of $Li_2S$, LiCl, and/or $Li_3PO_4$) into vials which are quite toxic and aggressive to current collectors and lithium metal anode sources.

increases interfacial resistance between lithium and the solid electrolyte, which hampers lithium-ion transport and consequently reduces battery performance. Furthermore, the $Li_2S_2$ compound was also identified in the Li 1s and S 2p spectra of the Li metal current collector, resulting from sulfide electrolyte decomposition. In the P 2p spectrum, a small amount of $P_xO_y$ species is detected at higher binding energy around 140.0 eV, which was observed in nearly all current collectors[54]. Lastly, the Cu/Li current collector exhibited some electrolyte degradation products on the surface of Li (Fig. 3g). Like the case with SS, a trace of the LiCl compound was found at 56.5 eV. In addition to metallic Li, a peak related to $Li_2O$ was observed at 55.2 eV in the Li 1s spectrum[55]. This appearance is also noted in the O 1s spectrum of the Cu/Li current collector at a binding energy of 529.5 eV. The formation of $Li_2O$ indicates an interaction between lithium and impurities or electrolytes. The identification of the $Li_2O$ phase is in good agreement with the XRD data (Fig. 1g).

XPS spectra of the current collectors, emphasizing the Cu, Fe, Ni, and Al 2p regions shown in Fig. 4. As mentioned about the Cu collector clearly showed strong peaks of $Cu_2S$ in Cu 2p region (Fig. 4a) at binding energy of 933.0 (2p3/2) and 952.8 eV (2p1/2)[48]. Fig. 4b shows the high resolution XPS spectrum of the SS current collector. It can be seen that there are three peaks in binding energy region at 707.0, 710.5, and 712.2 eV, which can be assigned to Fe 2p3/2 for Fe, FeO, and $Fe_2O_3$, respectively[56,57]. The position of these primary peaks is consistent with that of the core-level XPS spectrum of metallic Fe, FeO and $Fe_2O_3$. The higher binding energy peaks correspond to Fe 2p1/2 for the same compounds of SS. Figure 4c shows the high-resolution Ni 2p XPS of Ni. Three main peaks located at 852.3, 853.3, and 856.0 eV were observed for Ni 2p3/2 of Ni, NiO, and $Ni_2O_3$, respectively[57]. The satellite peak at around 880.2 eV and 861.6 eV are two shake-up type peaks of nickel at the high binding energy side of the Ni 2p1/2 and Ni 2p3/2 edge. The satellite peaks usually appear when there is an unpaired electron in the metal 3 d orbital. As it can be seen in Fig. 4c, two peaks corresponding to $A_2O_3$ and Al are present at 74.5 and 71.6 eV in the Al 2p region of Al current collector[58]. These metal oxides peaks were also observed in the O 1 s regions (Fig. 3a–g). This indicated that the surface of SS, Ni, and Al current collector as oxidized before XPS measurement.

It should be noted that most experiments with $Li_6PS_5Cl$ electrolytes are conducted in a Glove Box at water and oxygen <0.1 ppm, however, the high chemical reactivity of sulfur to metal surfaces is still an open question from the point of view of chemical corrosion[26,32,33,59,60]. Therefore, in order to investigate the chemical stability of current collectors and lithium metal sources (pristine Li-foil and bi-layer Cu/Li-foil metal anodes), side reaction components were formed on their surface in non-direct contact for 2 months, as shown in Fig. 5. This experiment simulates the interaction between the LPSCl pellet and the current collectors and lithium metal anode sources, leading to the potential formation of side reaction products such as $Li_3P$, $Li_2S$, LiCl and/or $Li_3PO_4$ and byproducts depending on the type of current collectors (aforementioned above). These impurity phases are toxic and can aggressively react with the current collectors and lithium metal anodes. Understanding the chemical stability and reactivity of these current collectors when exposed to the LPSCl electrolyte is crucial for assessing their suitability in lithium metal solid-state batteries with sulfide solid electrolyte. The findings provide valuable insights into the degradation mechanisms and help identify the most stable and efficient materials for use in solid-state battery systems. At 60 °C, current collectors (SS, Ni, Cu, Al, Al/C) are susceptible to thermal expansion, increased corrosion, and elevated electrical resistance. In the case of lithium metal anodes (Li, Cu/Li), elevated temperatures can facilitate dendrite growth, electrolyte decomposition, and interfacial instability. Therefore, elevated temperatures can significantly impact chemical decomposition and structural integrity of the current collectors and lithium metal anodes. As certified by photos with direct contact in LPSCl electrolyte (Fig. 1) and XRD analysis (Fig. 2), the surface changes of the samples were found by the treatment and indicated unfavourable thermodynamic instability of the current collectors and lithium metal sources with the formation of undesirable side reaction components leading to chemical corrosion of them. In addition, the surface of the SS, Ni, Al and Al/C collectors maintained a high degree of stability compared to the Cu with formed corroded areas and a similar effect is observed for the pure Li and Cu/Li metal anodes (Fig. 5c, g). The colour of the Cu, pristine Li metal foil and Cu/Li foil metal anodes is also a good indicator of their instability in contact with the LPSCl electrolyte (Fig. 1), in agreement with previous reports[30,32,33,59] after air exposure and electrolyte treatment, for example, but

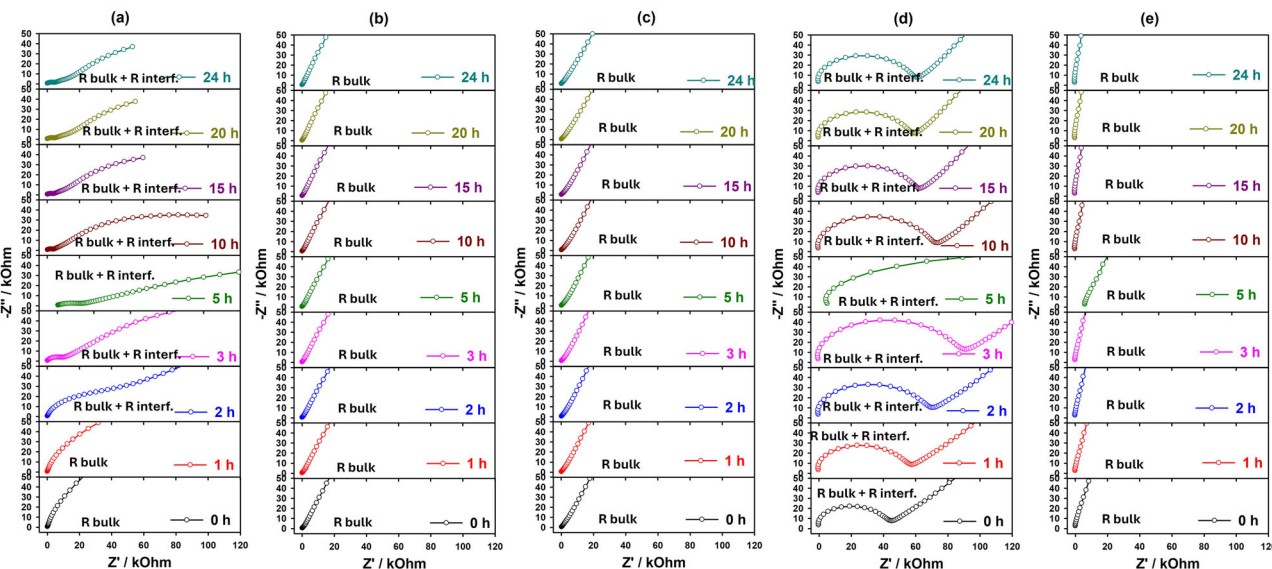

**Fig. 6 | EIS spectra of current collectors before and after treatment with LPSCl.** EIS spectrums current collector of **a** Cu, **b** SS, **c** Ni, **d** Al, and **e** Al/C before and after direct contact with LPSCl electrolyte for 24 h.

in our cases there are formed side reaction components compared to the electrolyte treatment with new SEI layers formed consists of byproducts of lithium metal anode, current collectors and sulfide solid electrolyte.

Big efforts have been dedicated to understand the behavior of sulfide-based electrolytes when are in contact with cathode materials or lithium metal anodes. However, the use of current collectors is still an open question, especially from the point of view of chemical, electrochemical, thermal and mechanical failure. Therefore, experimental results of works such as electrochemical analysis have been carried out to understand the stability or instability of current collectors and lithium metal anodes for solid-state batteries with $Li_6PS_5Cl$ sulfide-based electrolytes. To understand the cause of the electrochemical failure, the Nyquist plots for 24 h were obtained for Cu, SS, Ni, Al and Al/C current collectors in contact with the LPSCl electrolyte, as shown in Fig. 6 and Fig. S7 (Supplementary Information). It should be noted that the resistance obtained for SS, Ni and Al/C current collectors is stable and related to the total (bulk + grain boundaries (interface)) solid electrolyte resistance (Fig. 6b, c, e). About the Al collector, the interfacial resistance slightly increased for 24 h, it is still stable against LPSCl electrolyte on the subject of chemical reactivity between sulfide and current collector, but the tiny and undetectable amount of side products components could be possible (Fig. 6d). About the Cu current collector, the interfacial resistance moderately grew due to the formation of unwanted components on the Cu surface which are associated with poor chemical stability analysis (Figs. 1 and 2). Extending the contact time between the LPSCl and current collectors to over 24 h demonstrates a stable interfacial resistance, in contrast to that observed with Cu. This stability is attributed to the formation of side reaction products, such as the $Li_3P$ alloy, which forms a layer characterized by high electronic conductivity and low ionic conductivity, unlike the highly conductive layers formed by $Li_2S$, LiCl, and $Li_3PO_4$ components[30,61]. Consequently, the stabilization of interfacial resistivity for the Cu current collector, combined with the appearance of fully reduced reaction products ($Li_3P$, $Li_2S$, LiCl, and/or $Li_3PO_4$), as shown in Fig. 6. Thus, the formation of a more uneven and inhomogeneous SEI layer from interfacial products can be strongly suggested by the stabilization of the interfacial resistivity for the Cu current collector combined with the appearance of fully reduced reaction products ($Li_3P$, $Li_2S$, LiCl and/or $Li_3PO_4$) as presented in Fig. 6a. Furthermore, to understand the electrochemical behaviour of the current collectors concerning the LPSCl electrolyte, time and current ($i$–$t$) measurements were carried out for 24 h, taking into account that the voltage conditions are slightly different from the actual conditions of the lithium-ion and solid-state batteries (where the

sweep potential does not apply), as shown in Fig. S8 (Supplementary Information). Under the application of 4.3 V, the obtained time and current ($i$–$t$) results for SS, Ni, Al and Al/C current collectors maintain a highly stabilized current. However, for Cu current collectors, there was an increase in current for 24 h, which may be an indication of chemical and electrochemical reactivity—Cu corrosion in contact with the LPSCl electrolyte. The data obtained can be used to determine the faster Cu corrosion after long storage, similar to the actual conditions found in solid-state batteries[30,32]. After several hours of indirect or undirect contact between various current collectors (SS, Cu, Ni, Al, Al/C, Li, and Cu/Li) and the $Li_6PS_5Cl$ sulfide solid electrolyte, changes in resistance were observed due to several factors such as chemical reactions between the current collectors and the LPSCl electrolyte led to the formation of impurity phases, which deposited on the surface and increased resistance. Surface degradation, particularly in Cu, Li and Cu/Li, further impacted conductivity. Additionally, the formation of a solid electrolyte interphase (SEI) layer created a barrier to electron flow, contributing to increased resistance. Materials like SS, Ni, Al and Al/C maintained higher stability compared to Cu, Li and Cu/Li which showed more pronounced resistance changes due to their higher chemical reactivity. These observations highlight the importance of selecting chemically stable materials for current collectors in sulfide solid-state battery systems.

It should be considered that from the point of view of low cost and weight, chemical and electrochemical stability (corrosion stability against liquid electrolyte or/and ionic liquid components), common metals (Al, Cu, Ni and stainless steel) should be replaced in the future by eco-friendly. chemically stable and light materials[13,15,17,21]. On the other hand, in the case of solid-state batteries with sulfide electrolytes, current collectors such as stainless steel foils are used and the investigation of different types of current collectors is still open, especially for the manufacture of cells from coin cell in press device format to pouch[31,62,63]. Thus, in this work, the SS, Cu and Ni current collectors were used as potential candidates for sulfide electrolytes in solid-state batteries with reversible lithium plating and stripping (Fig. S9, Supplementary Information). The cycling performance of SS, Cu and Ni current collectors in CC|LPSCl|Li/CC and CC/Li|LPSCl|Li/CC cells have been investigated by galvanostatic charge/discharge process by using a current density of 0.15 mA cm$^{-2}$ with a fixed capacity of 1 mAh cm$^{-1}$ as shown in Fig. 7a, b, respectively. The CC|LPSCl|Li/CC non-symmetric cell can sustain poor cycling and the Li plating/stripping voltage drops suddenly, indicating the occurrence of short-circuit after 20 h for all collectors (Fig. 7a). This aspect means that the Li dendrite formed and grew continuously in the non-symmetric cell during the cycling and has formed a

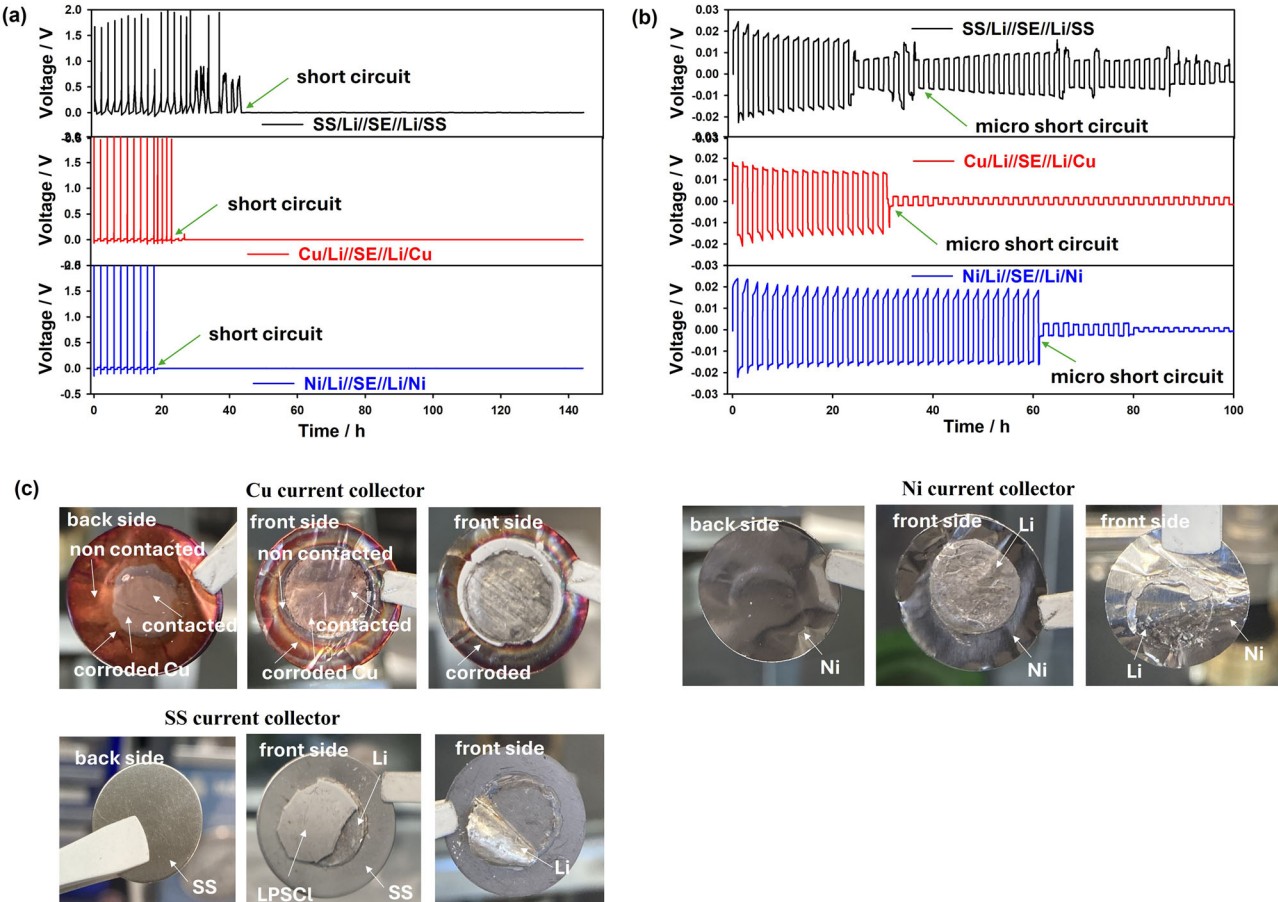

**Fig. 7 | Lithium plating experiments using different current collectors.** Lithium plating at a current density of 0.15 mA cm$^{-2}$ of current collectors of Cu, SS and Ni in **a** CC|LPSCl|Li/CC and **b** CC/Li|LPSCl|Li/CC cell formats, and **c** photos of current collectors, LPSCl pellets and lithium metal anode after plating. Note: back side of current collector after LPSCl solid electrolyte contact; one photo of front side shows the surface of current collector after removing the Lithium metal anode; second photo of front side shows together current collector and Lithium metal anode.

non-uniform layer, which leaded to a short circuit. On the other hand, as shown in Fig. 7c, it may be related to the corrosion of the Cu current collectors in contact with the LPSCl sulfide electrolyte. In contrast, the symmetrical cell of CC/Li|LPSCl|Li/CC shows more stable voltage profiles (Fig. 7b) compared to the non-symmetrical cell (Fig. 7a). However, the Cu collector short-circuit phenomenon is observed after 30 h, indicating that Li dendrite formation is significantly increased due to the reduced practical current density resulting from high surface area and corrosion effect. In addition, another reason related to the short circuit may be induced by the unstable interfacial boundary between the lithium metal anode and LPSCl solid electrolyte and electrical disconnection due to repeated growth/corrosion of Li dendrites, especially in the coin cell format where the internal pressure of 0.1 MPa exceeds that in the press device at operating pressure of over 1 MPa[26,27]. Compared to Cu, SS and Ni foils maintain a more stable cycling performance, which may be related to the stable interfacial boundary and the inhibition of dendritic Li formation. However, due to the still poor internal contact in the coin cell configuration cell between the lithium metal anode and the solid electrolyte, the short-circuit phenomena are observed and change the voltage hysteresis curves (Fig. 7b). While Ni and SS collectors maintain stable chemical and electrochemical properties against LPSCl solid electrolyte after cycling without visible corrosion compared to Cu foil (Fig. 7c). By considering that back side of Cu change color, it's possible to assume that liquid polysulfides could be formed during cycling being able to oxidize Cu to Cu$_2$S/CuS. It was observed that the SS and Ni foils have excellent contact between the LPSCl solid electrolyte and the current collectors with a similar surface morphology to the plated Li without any

corrosion effects. For Cu foil, it was observed that the corroded Cu current collector was in direct contact with the LPSCl solid electrolyte and in indirect contact, which is associated with poor chemical stability (undirect contact in vials, Fig. 5). Furthermore, the high overpotential and voltage hysteresis can be attributed to the large internal resistance due to the accumulation of dead Li and the formed side reaction components (Fig. 7). Moreover, it should be noted that in the CC | LPSCl | Li/CC system, lithium ions from the Li$_6$PS$_5$Cl sulfide solid electrolyte are reduced and deposited onto the copper current collector (for example) during the plating process. During stripping, these lithium ions are oxidized back into the solid electrolyte. This system faces challenges such as dendrite formation and uneven stripping, which can lead to capacity loss and short circuits. In contrast, the CC/Li|LPSCl|Li/CC system involves the direct deposition of lithium ions onto the lithium metal anode during plating. During stripping, lithium metal is oxidized back into ions that migrate into the solid electrolyte. This system also encounters issues with dendrite growth and the formation of nanovoids, which can reduce cycling efficiency. The primary differences between these systems lie in the current collector used for lithium deposition and the associated challenges, highlighting the importance of current collector selection in optimizing the performance and safety of solid-state batteries. Thus, it could be concluded that copper and nickel generally offer better plating and stripping results for lithium metal compared to stainless steel due to several key factors. These include the formation of easy alloys on the surface of copper and nickel, higher thermal conductivity, and greater ductility, which facilitate efficient heat transfer and smoother finishes. Additionally, copper and nickel have better chemical compatibility with lithium metal anodes,

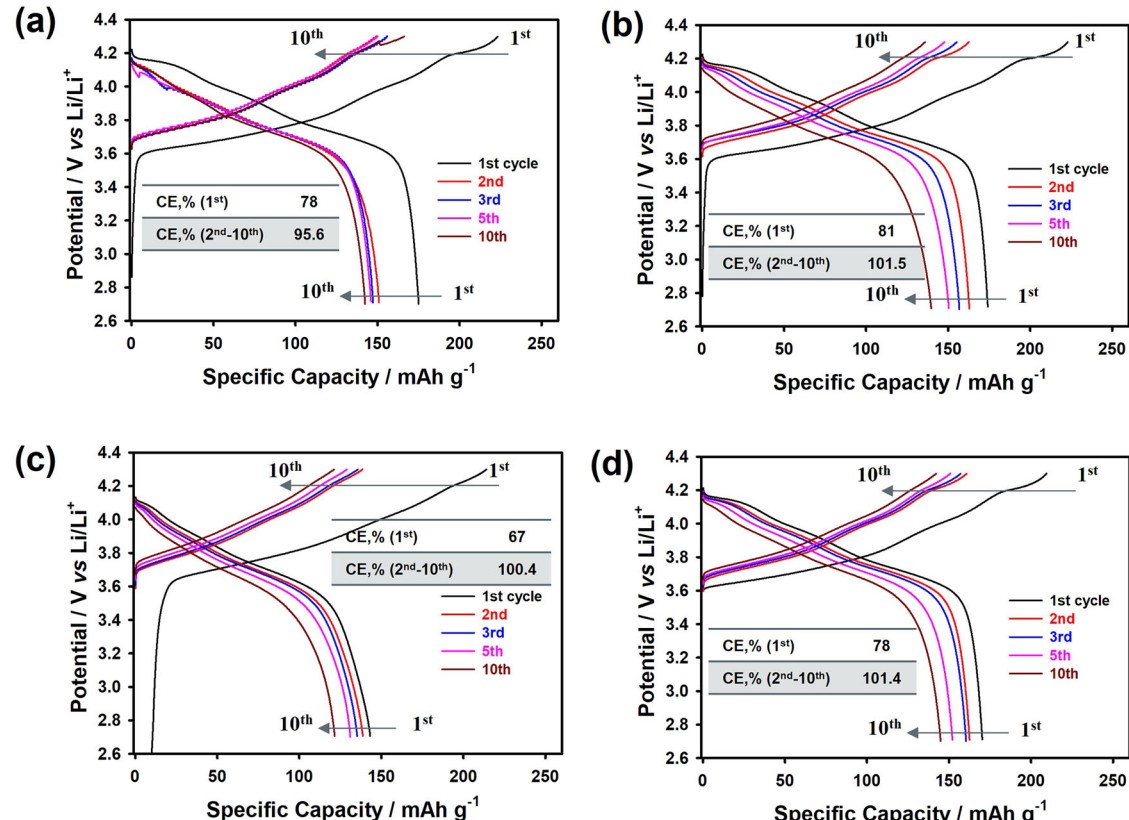

**Fig. 8 | Cycle life of NCM90505|LPSCl|Li/SS cells using different current collectors.** Cycle-life of **a** NCM90505|LPSCl|Li/SS cell, **b** NCM90505|LPSCl|Li/Ni cell, **c** NCM90505|LPSCl|Li/Cu cell, and **d** NCM90505|LPSCl|Li40/Cu (thickness of lithium on Cu foil is 40 μm) cell in a potential range of 2.7 and 4.3 V at room temperature at C/20 into coin cell format CR2032.

allowing for plating and stripping without damaging the underlying material. However, stainless steel, despite its hardness, provides superior chemical and electrochemical stability for long cycle life applications. Therefore, for high-quality and efficient plating and stripping processes of lithium, copper and nickel alloys are preferred, while stainless steel is better suited for applications requiring long-term stability.

To reduce the influence of NCM cathodes and to better compare lithium anode performance in contact with collectors, the effect of collectors on half cells over cycle life is shown in Fig. 8. The NCM90505 | LPSCl|Li/SS half cell (Figs. S10 and S11, Supplementary Information) shows much better cycle stability and lower polarization, indicating high Li metal utilization in contact with the SS collector (Fig. 8a). In contrast, the Ni, Cu and Cu/Li-40 μm half cells have lower specific capacity and much higher polarization with high internal resistance even after 10 cycles (Fig. S12, Supplementary Information), showing abrupt capacity drops due to dead Li formation during cycling life and low Li utilization in contact with these collectors compared to the Li/SS anode (Fig. 8). Regarding the Coulombic efficiency for the first cycle, it is prominent that the SS current collector greatly improves the initial Coulombic efficiency compared to Ni, Cu and Cu/Li-40 μm, which are 78%, 67%, 81% and 78%, respectively. While, the average Coulombic efficiencies after 10 cycles of the cells were 95.6%, 100.4%, 101.5% and 101.4% for SS, Ni, Cu and Cu/Li-40 μm respectively. The values obtained for the SS current collectors are lower than the average Coulombic efficiency of the Ni, Cu and Cu/Li samples[64]. These results also confirm that the SS collectors are effective in the reduction of irreversible reactions between current collectors and LPSCl solid electrolyte during the first cycle. Thus, it was confirmed that the capacity obtained during charging was completely reversible during the subsequent discharge for SS current collectors, and the electrolyte side reaction appeared to be successfully reduced. In contrast, for Ni, Cu and Cu/Li 40 μm, the formation of side reaction

components was observed, leading to capacity fading of the cells. It should be noted that the half cell with Li/SS shows a lower polarization, which can be attributed to the advantages of the SS current collectors that Cu and Ni, thanks to the stable interfacial contact and the low internal resistance (Fig. S12, Supplementary Information) and more chemical and electrochemical stabilities against more aggressive LPSCl sulfide solid electrolytes that are associated with the obtained abovementioned results. Thus, we can conclude that the promising current collectors based on SS are the best candidates for solid-state batteries with LPSCl sulfide solid electrolytes thanks to the high chemical and electrochemical stabilities, the SS can partly accommodate the volume variation during Li plating/stripping prolong the growth of lithium dendrite with reducing the voltage hysteresis. The superior performance of NCM cathodes with LPSCl sulfide solid electrolyte and lithium metal anode in contact with stainless steel, compared to copper and nickel current collectors, is primarily due to stainless steel's better chemical and electrochemical stability, excellent corrosion resistance, high mechanical strength, and compatibility with sulfide electrolytes. These properties collectively enhance the interface stability and overall battery performance, making stainless steel a more suitable choice for long-term applications. Moreover, non-uniform solid electrolyte films with contact lithium metal anode can form lithium dendrites depositing preferentially in the interior of avoids and cracks of solid electrolyte that lead to the capacity fading of cells. Therefore, to reach stable electrochemical performance, the uniform lithium nucleation and deposition must be regulated with a homogenous distribution of lithium-ion flux on the whole surface of a solid electrolyte into the coin cell format or press devise with the high precision distribution of area pressure. It is still an open question related to the internal pressure in coin cell format compared to the press device if the manufacturing of the sulfide solid-state battery on a large scale in needs to optimise and improve the sample holder with low pressure maintaining the

stable cycling life, however, it is so far from the industrialization of these type of cells.

## Conclusions

In this work, systematic physicochemical and electrochemical analyses were carried out on various current collectors made of copper, nickel, stainless steel, aluminum and aluminum carbon in contact with the $Li_6PS_5Cl$ electrolyte in a coin cell configuration, to understand and investigate the reaction mechanisms. It should be noted that SS, Ni, Al and Al/C show good chemical and electrochemical compatibility for long-term contact, whereas Cu, Li and Cu/Li have a high susceptibility to corrosion in sulfide electrolytes with the possible formation of side reactions such as CuS, $Li_2O$ and CuP, respectively, and byproducts between lithium metal anode sources, current collectors and $Li_6PS_5Cl$ sulfide solid electrolytes. Moreover, selecting the appropriate current collector for sulfide solid-state batteries is crucial for ensuring stable long-term performance. This choice depends on several key factors, including chemical and electrochemical reactions (formation of side reaction components and byproducts), surface degradation (corrosion impact), solid electrolyte interphase (SEI) layer formation, and material stability (compatibility with the sulfide solid electrolyte). Therefore, these results provide the needed understanding for the current collector's selection in sulfide component fabrication, especially via wet chemistry, which is a promising approach for the industrial development of sulfide electrolyte solid-state batteries.

## Data availability

The data supporting this study's findings are available from https://doi.org/10.6084/m9.figshare.29255948.v1 (All raw data are available in Supplementary Data File 1).

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

## Acknowledgements

This work was supported by the financial support of the Austrian Federal Ministry for Climate Action, Environment, Energy, Mobility, Innovation, and Technology. The presented work was supported by the European Union's Horizon Europe programme for research and innovation under grant agreement No. 101069686 (PULSELION project). Furthermore, the authors want to express their gratitude to Raad Hamid for the XRD and XRF measurements of samples.

## Author contributions

Conceptualization: A.T., A.B., Methodology: A.T., A.B., General Experimental investigation and electrochemical analysis: A.T., A.B., Characterization XRD: A.T., I.M., Characterization XPS and XRF: I.M., Writing—original draft preparation: A.T., A.B., Writing—review and editing: A.T., A.B., I.M., A.P.

## Competing interests

The authors declare no competing interests.
