## [Transparent Peer Review file · Communications Chemistry]

Probing the chemical stability between current collectors and argyrodite Li₆PS₅Cl sulfide electrolyte

Corresponding Author: Dr Artur Tron

Version 0:

Reviewer comments:

Reviewer #1

(Remarks to the Author)

The study explores the stability of sulfide solid electrolytes with various types of current collectors. This research has the potential to contribute significantly to the development of chemically and electrochemically stable current collectors, a crucial factor for advancing anode-free all-solid-state batteries. While the study presents an interesting concept and methodology, several findings appear contradictory, and the overall analysis lacks sufficient depth. Incorporating additional characterization techniques such as XPS, TEM, and CV could enhance the study's impact and make it more appealing to the readers of Communications Chemistry. Below are my detailed comments and suggestions for the authors:

1. The Nyquist plot highlights substantial changes in resistance growth over time for Cu and Al current collectors. What explanation do the authors provide for the observed behavior of Al/CC? Specifically, the impedance increases up to 3 hours and then decreases. Is this fluctuation attributable to interfacial reactions, electrolyte degradation, or another mechanism?
2. The XRD patterns and optical images indicate that stainless steel (SS) exhibits better stability compared to other current collectors. However, these results contradict the findings from the symmetric tests. For instance, in Figure 5(b), Cu and Ni demonstrate better cycle life in the CC/Li|LPSCI|Li/CC configuration. How do the authors reconcile this inconsistency?
3. In the NCM90505|LPSCI|Li/SS cell, better cyclability with lower polarization is observed compared to NCM90505|LPSCI|Li/Ni and NCM90505|LPSCI | Li/Cu cells. However, these results are inconsistent with the half-cell and symmetric cell trends shown in Figure 5.
 - (a) What causes the SS|LPSCI|Li/SS and SS/Li|LPSCI|Li/SS cells to experience short-circuiting and other disruptions, unlike other configurations?
 - (b) Why does the NCM90505|LPSCI | Li/Ni cell exhibit high polarization despite its stable symmetric cell cyclability? Similarly, how can the behavior of the NCM90505|LPSCI|Li/SS cell be explained?
4. To better understand the interaction between LPSCI and different current collectors, can the authors investigate changes in the crystalline structure of the particles upon contact? Conducting TEM measurements would provide valuable insights.
5. The study discusses the formation of various side products resulting from the reaction between LPSCI and current collectors such as Cu, Li, and Al/C. To substantiate this claim, the authors should consider using XPS or other highly accurate analytical techniques.
6. I recommend performing cyclic voltammetry measurements with the current collectors under study and providing a comparative analysis.
7. The formation of Li₂O is mentioned in the manuscript. Where does this come from?

Reviewer #2

(Remarks to the Author)

This paper studies the chemical stability between various current collectors and argyrodite Li₆PS₅Cl sulfide electrolyte. The authors of this paper studied the chemical stability of the current collector, which has not received much attention in sulfide-based all-solid-state batteries. However, I would like to request a major revision as I believe the analysis and evidence need to be greatly strengthened compared to the topic of the paper.

The items that need to be corrected and supplemented are as follows.

In line 59, this paper is about chemical reactions with the entire body. Please describe undesirable side reaction components specifically without relying on references and provide the basis for it.

The numbers of figures and supplementary information do not correspond in order to the flow of the text in the main text, but rather go back and forth, which is inconvenient for the reader. For example, in line 74, it starts from figures S7, which is awkward. The numbers of figures and supplementary information figures do not correspond in order to the flow of the text in the main text, but rather go back and forth, which makes the reader uncomfortable.

In line 82, please do not lump Figure S1 and Figure S2 together and describe them, but clearly describe what you are trying

to explain about each.

In line 86, explain for the reader why the untreated sample is untreated.

From line 91 to line 97, it is difficult to assert that the effect of impurities is simply due to low density. An explanation is needed in conjunction with the impurity analysis results of XRD structural analysis.

From the beginning of the text to line 97, I am not sure if it is really necessary at the beginning of the text. Since it is mostly about experimental conditions, it seems that most of it can be removed as experimental or supplementary information.

In the experimental conditions, why was 2 months chosen instead of 1 month or 3 months or more?

Most of the electrochemical evaluations of all-solid-state batteries are conducted at high temperatures above room temperature. You can see the deterioration due to the storage time, but the chemical changes with the current collector due to temperature changes also seem to be important. Is it possible to add more?

In line 100, why should figure S2 be viewed together with figure 1? Please explain in detail why each is necessary.

This is the caption part of figure 1(g) in line 120.

The front and back of the entire Cu collector should be described, but the table in the figure describes them as pristine and corroded. It seems that the description and the table do not match.

In line 137, the abbreviation 'didn't' is not appropriate for the written language of a research paper.

Figures 2, 4, 5, 6 and figures S7, S9 are too small to read. You should divide the figures into larger ones or increase the font size. Published papers will be read by readers in various environments. Considering this, authors should print them out and check that they are readable.

It is necessary to present a possible chemical reaction formula and its basis for the byproduct in Figure 2 that is generated by the side reaction between the sulfide solid electrolyte and the current collector.

A detailed explanation is needed as to why the lithium deposition pattern is different in figure 5(c).

In lines 39-44 of the introduction, the possible reactions of the solid electrolyte need to be specified and additional references need to be provided.

It is difficult to distinguish between the front side/back side/LPSCI in the 'after 24h' pictures in Figure 1. Please indicate them in each picture.

In figure 1 (f) and (g), it needs to be precise and concise as to whether it is 2 months or 3 months. (Line 102 of the text indicates 2 months, and the caption of Figure 1 (f) says 3 months.)

In figure 1 (a) 'after 24h', a noticeable surface change is observed, but why are no impurities observed in the XRD data of figure 2 (a)?

Why is 'after 1 week data' included only in figure 2 (a), and why are there no after 1 week data in (b)-(g)?

There is an XRD analysis result in Figure 2, but why are there no pictures of Li and Cu/Li in Figure 1?

I would like to mark each picture in Figure 3 with alphabet symbols such as (a), (b)...(i)... and provide a related explanation.

In all the samples in Figure 4, only the 5h part seems to deviate from the trend. Additional explanation is needed for this data.

The number of cycles (10 times) in Figure 6 is too short. This study is about the long-term chemical stability of the current collector and solid electrolyte for more than 3 months, so it is necessary to present long-life cycle data accordingly.

Reviewer #3

(Remarks to the Author)

In this manuscript, the authors reported the systematic analyses of interfaces between current collectors and solid electrolyte. Before the acceptance of the manuscript, major revisions should be made, and the following questions need to be further considered.

Although this work is to understand and investigate the reaction mechanism of the interface of LPSCI and current collectors, the reaction products have not been identified and then the reaction mechanism has not been clear. XPS analysis helps to understand the decomposition products and reaction mechanisms.

In Line 71, is "scarring" mistaken for "screening"?

In Line 77, the correct space group is "F-43m".

In Fig. 1, why is there the non-contact area in the cell using Cu foil? Does the current collector and solid electrolyte not have the same size?

There are two abbreviations for Li₆PS₅Cl, LPSC and LPSCI (e.g., Fig. 5 (c), Fig. S10 (a), Fig. S8, Fig. 2, and line 205).

Version 1:

Reviewer comments:

Reviewer #1

(Remarks to the Author)

The authors have addressed all comments well, and the quality of the revised manuscript has been substantially improved. It may be suitable for publication in Communications Chemistry

15.04.2025

Reference: Revised Manuscript Submission to Communications Chemistry

We thank you for the opportunity to submit a revised version of our manuscript. We are grateful to all reviewers for their efforts to revise our manuscript and sincerely value their thoughtful and constructive suggestions. The details of the revisions made in the revised manuscript along with our response to the reviewer comments are as following, where the changes to our original manuscript are **highlighted in yellow**.

Reviewer #1 (Remarks to the Author):

The study explores the stability of sulfide solid electrolytes with various types of current collectors. This research has the potential to contribute significantly to the development of chemically and electrochemically stable current collectors, a crucial factor for advancing anode-free all-solid-state batteries. While the study presents an interesting concept and methodology, several findings appear contradictory, and the overall analysis lacks sufficient depth. Incorporating additional characterization techniques such as XPS, TEM, and CV could enhance the study's impact and make it more appealing to the readers of Communications Chemistry. Below are my detailed comments and suggestions for the authors:

1. The Nyquist plot highlights substantial changes in resistance growth over time for Cu and Al current collectors. What explanation do the authors provide for the observed behavior of Al/CC? Specifically, the impedance increases up to 3 hours and then

decreases. Is this fluctuation attributable to interfacial reactions, electrolyte degradation, or another mechanism?

Thanks for your valuable comment. We can suppose that after extending the contact time with the LPSC and current collectors to more than 24 hours demonstrates a stable interfacial resistance compared to Cu, for example. This stability is attributed to the formation of side reaction components, such as the Li₃P alloy, which creates a layer with high electronic and low ionic conductivities, unlike the highly conductive layers formed by Li₂S, LiCl, and Li₃PO₄ components [1,2]. Thus, the formation of a more uneven and inhomogeneous SEI layer from interfacial products can be strongly suggested by the stabilization of the interfacial resistivity for the Cu current collector combined with the appearance of fully reduced reaction products (Li₃P, Li₂S, LiCl and/or Li₃PO₄) as presented in Figure 4a. Therefore, we added some explanations into a body of text which you can find below.

[1] A. Tron, A. Beutl, I. Mohammad, A. Paoella, Insights into the chemical and electrochemical behavior of halide and sulfide electrolytes in all-solid-state batteries, *Energy Advances* (2025).

[2] D.H.S. Tan, E.A. Wu, H. Nguyen, Z. Chen, M.A.T. Marple, J.M. Doux, X. Wang, H. Yang, A. Banerjee, Y.S Meng, Elucidating reversible electrochemical redox of Li₆PS₅Cl solid electrolyte. *ACS Energy Lett.* 4 (2019) 2418–2427.

(original version, page 13, in the Results and Discussion section)

Extending the contact time between the LPSCl and current collectors to over 24 hours demonstrates a stable interfacial resistance, in contrast to that observed with Cu. This stability is attributed to the formation of side reaction products, such as the Li₃P alloy, which forms a layer characterized by high electronic conductivity and low ionic conductivity, unlike the highly conductive layers formed by Li₂S, LiCl, and Li₃PO₄ components [30,62]. Consequently, the stabilization of interfacial resistivity for the Cu current collector, combined with the appearance of fully reduced reaction products (Li₃P, Li₂S, LiCl, and/or Li₃PO₄), as shown in Figure 6.

(original version, page 8, in the Results and Discussion section)

[30] D.H.S. Tan, E.A. Wu, H. Nguyen, Z. Chen, M.A.T. Marple, J.M. Doux, X. Wang, H. Yang, A. Banerjee, Y.S. Meng, Elucidating reversible electrochemical redox of $\text{Li}_6\text{PS}_5\text{Cl}$ solid electrolyte. *ACS Energy Lett.* 4 (2019) 2418–2427.

[62] A. Tron, A. Beutl, I. Mohammad, A. Paoletta, Insights into the chemical and electrochemical behavior of halide and sulfide electrolytes in all-solid-state batteries, *Energy Advances* (2025).

2. The XRD patterns and optical images indicate that stainless steel (SS) exhibits better stability compared to other current collectors. However, these results contradict the findings from the symmetric tests. For instance, in Figure 5(b), Cu and Ni demonstrate better cycle life in the CC/Li|LPSCI|Li/CC configuration. How do the authors reconcile this inconsistency?

Thanks for your comment. Copper and nickel generally offer better plating and stripping of lithium metal results compared to stainless steel due to several key factors such as the formation of easy alloys on the surface of copper and nickel than stainless steel due to surface resistance, and thermal conductivity, for example, copper and nickel have higher thermal conductivity compared to stainless steel, which can be beneficial in plating/stripping processes that require efficient heat transfer, for example. Copper and nickel are more ductile than stainless steel, making them easier to work with during plating and stripping processes. This flexibility allows for better adherence and smoother finishes. Another one is chemical compatibility with lithium metal anode for plating/stripping of lithium for a short time of process without damaging the underlying material compared to stainless steel which has more hardness. Therefore, for plating/stripping process of lithium of copper and nickel alloy, for example, requires high quality and efficiency for the application of these materials. However, from the point of view of chemical and electrochemical stability, stainless steel is a much better material for long cycle life. We added this point into manuscript below.

(original version, page 15, in the Results and Discussion section)

Thus, it could be concluded that copper and nickel generally offer better plating and stripping results for lithium metal compared to stainless steel due to several key factors. These include the formation of easy alloys on the surface of copper and nickel, higher thermal conductivity, and greater ductility, which facilitate efficient heat transfer and smoother finishes. Additionally, copper and nickel have better chemical compatibility with lithium metal anodes, allowing for plating and stripping without damaging the underlying material. However, stainless steel, despite its hardness, provides superior chemical and electrochemical stability for long cycle life applications. Therefore, for high-quality and efficient plating and stripping processes of lithium, copper and nickel alloys are preferred, while stainless steel is better suited for applications requiring long-term stability.

3. In the NCM90505|LPSCI|Li/SS cell, better cyclability with lower polarization is observed compared to NCM90505|LPSCI|Li/Ni and NCM90505|LPSCI | Li/Cu cells. However, these results are inconsistent with the half-cell and symmetric cell trends shown in Figure 5.

(a) What causes the SS|LPSCI|Li/SS and SS/Li|LPSCI|Li/SS cells to experience short-circuiting and other disruptions, unlike other configurations?

(b) Why does the NCM90505|LPSCI | Li/Ni cell exhibit high polarization despite its stable symmetric cell cyclability? Similarly, how can the behavior of the NCM90505|LPSCI|Li/SS cell be explained?

Thank you for your remark. We agree with your comments regarding the difference into electrochemical process with various current collectors. Some explanations you can find below as.

(original version, page 17, in the Results and Discussion section)

The superior performance of NCM cathodes with LPSCI sulfide solid electrolyte and lithium metal anode in contact with stainless steel, compared to copper and nickel current collectors, is primarily due to stainless steel's better chemical and electrochemical stability, excellent corrosion resistance, high mechanical strength, and compatibility with sulfide electrolytes.

These properties collectively enhance the interface stability and overall battery performance, making stainless steel a more suitable choice for long-term applications.

4. To better understand the interaction between LPSCl and different current collectors, can the authors investigate changes in the crystalline structure of the particles upon contact? Conducting TEM measurements would provide valuable insights.

Thank you for your suggestion. We consider your point and we found that more attractive to show the XPS analysis of the current collectors and lithium metal anode sources after contact with Li₆PS₅Cl with element analysis, where we can see the chemical stability of these samples and Li₆PS₅Cl solid electrolyte below in comment 5.

1

5. The study discusses the formation of various side products resulting from the reaction between LPSCl and current collectors such as Cu, Li, and Al/C. To substantiate this claim, the authors should consider using XPS or other highly accurate analytical techniques.

Appreciate your thoughts. We added the XPS analysis of the current collectors and lithium metal anode sources, which you can find below.

(original version, page 9-11, in the Results and Discussion section)

To explore the evolution of surface chemistry for different cycled current collectors, XPS spectra were acquired from the electrolyte-facing side of the current collectors after cycling. Figure 3 displays the XPS spectra of Cu, SS, Ni, Al, Al/C, Li, and Cu/Li, focusing on the Li 1s, S 2p, P 2p, and O 1s regions. The core XPS spectrum of the Cu current collector is shown in Figure 3a. As expected, no component of Li was detected in the Li 1s spectrum for Cu. The high-resolution S 2p spectrum for Cu reveals a significant doublet located at 161.7 eV (S 2p_{3/2}) and 162.9 eV (S 2p_{1/2}), attributed to the Cu₂S compound [49]. Confirmation of the presence of Cu₂S is also observed in the Cu 2p spectrum (Figure 4a) [50]. This finding aligns with the results from the XRD pattern (Figure 2a). The formation of Cu₂S typically occurs when the decomposed species S²⁻ from the Li₆PS₅Cl electrolyte reacts directly with Cu. Its formation is usually undesirable as it degrades the interface, leading to increased resistance and

mechanical issues (volume changes) at the interface. As seen in the Li 1s spectrum of the SS current collector (Figure 3b), several peaks related to lithium components were observed. One peak located at 56.0 eV corresponds to the LiCl compound [51]. A doublet at 198.5 eV ($2p_{3/2}$) and 200.0 eV ($2p_{1/2}$) is observed in the Cl 2p region, confirming the presence of LiCl (Figure S6a, Supplementary Information). Another peak in the Li 1s spectrum at 55 eV for SS is likely related to Li_2S_2 [52]. Additionally, traces of sulfite and P_2S_5 compounds are detected at 168.8 eV and 161.6 eV ($2p_{3/2}$) in the S 2p region, respectively. The XPS spectra of the Ni current collector show a trace of P_2S_5 in both the S 2p and P 2p regions at binding energies of 132.9 eV and 160.9 eV ($2p_{3/2}$) [53]. The Al and Al/C current collectors exhibit similar species, including sulfite and PO_4^{3-} in the S 2p and P 2p spectra, as depicted in Figures 3(d-e). For the Li metal current collector, Li_2CO_3 was identified in both the Li 1s and C 1s spectra (Figure S6b, Supplementary Information) at binding energies of 55.6 eV and 290.2 eV, respectively [54]. The formation of Li_2CO_3 on the surface of the lithium anode increases interfacial resistance between lithium and the solid electrolyte, which hampers lithium-ion transport and consequently reduces battery performance. Furthermore, the Li_2S_2 compound was also identified in the Li 1s and S 2p spectra of the Li metal current collector, resulting from sulfide electrolyte decomposition. In the P 2p spectrum, a small amount of P_xO_y species is detected at higher binding energy around 140.0 eV, which was observed in nearly all current collectors [55]. Lastly, the Cu/Li current collector exhibited some electrolyte degradation products on the surface of Li (Figure 3g). Like the case with SS, a trace of the LiCl compound was found at 56.5 eV. In addition to metallic Li, a peak related to Li_2O was observed at 55.2 eV in the Li 1s spectrum [56]. This appearance is also noted in the O 1s spectrum of the Cu/Li current collector at a binding energy of 529.5 eV. The formation of Li_2O indicates an interaction between lithium and impurities or electrolytes. The identification of the Li_2O phase is in good agreement with the XRD data (Figure 1g).

Figure 3. Deconvoluted XPS detail spectra for the Li 1s, S 2p, P 2p, and O 1s signals of (a) Cu, (b) SS, (c) Ni, (d) Al, (e) Al/C, (f) Li, and (g) Cu/Li current collectors.

XPS spectra of the current collectors, emphasizing the Cu, Fe, Ni, and Al 2p regions shown in Figure 4. As mentioned about the Cu collector clearly showed strong peaks of Cu_2S in Cu 2p region (Figure 4a) at binding energy of 933.0 (2p_{3/2}) and 952.8 eV (2p_{1/2}) [49]. Figure 4b shows the high resolution XPS spectrum of the SS current collector. It can be seen that there are three peaks in binding energy region at 707.0, 710.5, and 712.2 eV, which can be assigned to Fe 2p_{3/2} for Fe, FeO, and Fe_2O_3 , respectively [57,58]. The position of these primary peaks is consistent with that of the core-level XPS spectrum of metallic Fe, FeO and Fe_2O_3 . The higher binding energy peaks correspond to Fe 2p_{1/2} for the same compounds of SS. Figure 4c shows the high-resolution Ni 2p XPS of Ni. Three main peaks located at 852.3, 853.3, and 856.0 eV were observed for Ni 2p_{3/2} of Ni, NiO, and Ni_2O_3 , respectively [58]. The satellite peak at around 880.2 eV and 861.6 eV are two shake-up type peaks of nickel at the high binding energy side of the Ni 2p_{1/2} and Ni 2p_{3/2} edge. The satellite peaks usually appear when there is an unpaired electron in the metal 3d orbital. As it can be seen in Figure 4c, two peaks corresponding to Al_2O_3 and Al are present at 74.5 and 71.6 eV in the Al 2p region of Al current collector [59]. These metal oxides peaks were also observed in the O 1s regions (Figure 3a-g). This indicated that the surface of SS, Ni, and Al current collector as oxidized before XPS measurement.

Figure 4. The high-resolution deconvoluted XPS spectra of (a) Cu 2p for Cu, (b) Fe 2p for SS, (c) Ni 2p for Ni, and (d) Al 2p for Al current collectors.

6. I recommend performing cyclic voltammetry measurements with the current collectors under study and providing a comparative analysis.

Thanks for your viewpoint. We performed chronopotentiometry cyclic mode at a constant current density. This measurement can provide information on cell polarization, lithium deposition and dissolution kinetics, and dendrite propagation (Figure S7).

7. The formation of Li₂O is mentioned in the manuscript. Where does this come from?

Thanks for your valuable comment. The observed Li₂O phase on lithium metal in XRD peaks results from the formation of lithium oxide (Li₂O) during electrochemical reactions. Lithium oxide crystallizes in a cubic structure, producing distinct diffraction patterns detectable by XRD. This phase formation often occurs due to the interaction between lithium metal and

oxygen, leading to characteristic peaks in XRD analysis. Although, we used a dome for transporting and measuring samples into the Glove Box to avoid any contact with oxygen for XRD measurements.

We can suppose that it could be related to the simultaneously, native SEI on the surface of the lithium metal anode is conventionally described as having a multiphasic structure with fully reduced, dense ionic phases (such as Li_2O and/or LiF) closest to the Li interface in the "inner layer with the nanoscale SEI thickness, similar to the effects observed in lithium-ion battery systems with liquid electrolytes.

(original version, page 6, in the Results and Discussion section)

Moreover, the formation of Li_2O on the surface of the lithium metal anode after LPSCl contact may be attributed to the native SEI, which is conventionally described as having a multiphasic structure with fully reduced, dense ionic phases (such as Li_2O and/or LiF) closest to the Li interface in the "inner layer." This SEI exhibits nanoscale thickness, similar to the effects observed in lithium-ion battery systems with liquid organic electrolytes.

Reviewer #2 (Remarks to the Author):

This paper studies the chemical stability between various current collectors and argyrodite $\text{Li}_6\text{PS}_5\text{Cl}$ sulfide electrolyte. The authors of this paper studied the chemical stability of the current collector, which has not received much attention in sulfide-based all-solid-state batteries. However, I would like to request a major revision as I believe the analysis and evidence need to be greatly strengthened compared to the topic of the paper. The items that need to be corrected and supplemented are as follows.

1. In line 59, this paper is about chemical reactions with the entire body. Please describe undesirable side reaction components specifically without relying on references and provide the basis for it.

Thanks for your comment. Undesirable side reactions between copper (Cu) and the sulfide electrolyte $\text{Li}_6\text{PS}_5\text{Cl}$ can lead to the formation of various detrimental compounds. One significant issue is the potential formation of copper sulfides (Cu_xS / Cu_2S , CuS) due to the reaction between copper and the sulfur in the electrolyte. These copper sulfides can degrade the electrolyte's ionic conductivity and overall battery performance. Additionally, the interaction between copper and the electrolyte can result in the formation of phosphides (Cu_3P), which can further compromise the electrochemical stability of the system. These reactions are driven by the chemical reactivity of copper with the sulfur and phosphorus components of the $\text{Li}_6\text{PS}_5\text{Cl}$ electrolyte, leading to the formation of these undesirable phases. We added this point into body of paper.

(original version, page 2, in the Introduction section)

Previous works [31,32] have shown that solid sulfide electrolytes can react with Cu current collectors to produce undesirable side reaction components between copper (Cu) and the sulfide electrolyte $\text{Li}_6\text{PS}_5\text{Cl}$ can lead to the formation of copper sulfides (Cu_xS / Cu_2S and/or CuS) and phosphides (Cu_3P). These compounds degrade the electrolyte's ionic conductivity and compromise the electrochemical stability of the system.

2. The numbers of figures and supplementary information do not correspond in order to the flow of the text in the main text, but rather go back and forth, which is inconvenient for the reader.

Thank you for your insightful comment. We have taken your suggestion into account and have reorganized the figures and tables within the Supplementary Information.

3. For example, in line 74, it starts from figures S7, which is awkward. The numbers of figures and supplementary information figures do not correspond in order to the flow of the text in the main text, but rather go back and forth, which makes the reader uncomfortable.

Thank you for your valuable comment. We agree and have reorganized the Figures and Tables into the Supplementary Information.

4. In line 82, please do not lump Figure S1 and Figure S2 together and describe them, but clearly describe what you are trying to explain about each.

Thank you for your comment. We agreed and changed these sentences which you can find below.

(original version, page 3, in the Results and Discussion section)

The ionic conductivity of this solid electrolyte has been measured in pellet form, as illustrated in Figures S2 and S3 (Supplementary Information). Additionally, it has been evaluated when sandwiched between Li-ion blocking stainless steel (SS) disks in coin cell format CR2032 (Table S1, Supplementary Information), as shown in Figure S4 (Supplementary Information).

5. In line 86, explain for the reader why the untreated sample is untreated.

Thank you very much for your valuable comment. Untreated sample, we mean that this pristine powder of solid electrolyte which not undergo solvent treatment. We added this explanation into the body of text.

(original version, page 3, in the Results and Discussion section)

This results in higher impedance values and lower ionic conductivities compared to the untreated (pristine powder does not undergo solvent treatment) sample and is observed for solid electrolytes after solvent treatment [33,34].

6. From line 91 to line 97, it is difficult to assert that the effect of impurities is simply due to low density. An explanation is needed in conjunction with the impurity analysis results of XRD structural analysis.

Thanks for your valuable comment. In this section, we emphasize that while high ionic conductivity is essential for feasible battery operation, other parameters are equally important and have often been overlooked in the development of all-solid-state batteries. These parameters include the methods of fabricating solid electrolyte pellets and their measurements, such as ionic conductivity and critical current density. When lithium metal electrodes are used, breakdown is often initiated by significant dendrite formation and subsequent growth through the electrolyte layer, leading to internal shorting of the cell. However, the use of non-standardized cell setups and testing procedures makes it impossible to properly compare values. Therefore, we highlight that the impact of these often-overlooked parameters is significant in establishing more standardized measurement procedures for solid electrolytes under ambient or near-ambient pressure conditions.

(original version, page 3, in the Results and Discussion section)

This can be explained by differences in the mechanical properties of the obtained material from the theoretical value in terms of hardness and plasticity during densification by cold pressing, which can result in significant differences in the quality of the prepared powder pellets, and

consequently, samples with contaminants appear to have a higher degree of defects and porosity. This could be a further explanation for the slightly lower ionic conductivity and the type of method to be used for the measurement of ionic conductivity and density values [26,27,33,35].

7. From the beginning of the text to line 97, I am not sure if it is really necessary at the beginning of the text. Since it is mostly about experimental conditions, it seems that most of it can be removed as experimental or supplementary information.

Thank you for your valuable comment. Considering the chemical reactivity and stability of current collectors against sulfide solid electrolytes, it is crucial to identify materials that are suitable for both lab- and large-scale applications. This involves evaluating their long-term performance, durability, and overall cost-effectiveness. Additionally, selecting materials that are economically viable is essential to ensure the affordability and feasibility of cell production on a larger scale.

8. In the experimental conditions, why was 2 months chosen instead of 1 month or 3 months or more?

Thanks for your suggestion. We selected the 2-month period to demonstrate the visible corrosion effects of copper in contact with the sulfide solid electrolyte. However, we observed significant changes within 24 hours and one week, which greatly impact performance. Therefore, we included observations at the initial state, after 24 hours, and after 2 months to comprehensively understand the impact of the sulfide solid electrolyte on current collectors.

9. Most of the electrochemical evaluations of all-solid-state batteries are conducted at high temperatures above room temperature. You can see the deterioration due to the storage

time, but the chemical changes with the current collector due to temperature changes also seem to be important. Is it possible to add more?

Thanks for your valuable comment. We agree that elevated temperature has a huge impact on the performance. Therefore, we added some explanations and Figures of current collectors and lithium metal anode sources at 60°C, to understand the impact 60°C on samples for 4 hours which you find below.

(original version, page 12, in the Results and Discussion section)

At 60°C, current collectors (SS, Ni, Cu, Al, Al/C) are susceptible to thermal expansion, increased corrosion, and elevated electrical resistance. In the case of lithium metal anodes (Li, Cu/Li), elevated temperatures can facilitate dendrite growth, electrolyte decomposition, and interfacial instability. Therefore, elevated temperatures can significantly impact chemical decomposition and structural integrity of the current collectors and lithium metal anodes.

Figure 3. Photos of (a) SS, (b) Ni, (c) Cu, (d) Al, (e) Al/C, (f) Li and (g) Cu/Li current collectors after contact (undirect contact into vials) with LPSCl electrolyte after 2 months and after 4 hours heat treatment at 60°C. This experiment simulates the contact of LPSCl pellet and current collectors and the possible formation of side reaction components (impurity phases of Li_2S , LiCl , and/or Li_3PO_4) into vials which are quite toxic and aggressive to current collectors and lithium metal anode sources.

10. In line 100, why should Figure S2 be viewed together with Figure 1? Please explain in detail why each is necessary.

Thanks for your comment. To determine the chemical reactivity of current collectors (Cu, SS, Ni, Al, and Al/C) with the LPSCl sulfide electrolyte, we conducted experiments where these materials were in contact for 24 hours in a coin cell format (CR2032), as depicted in Figure 1 and Figure S4 (Supplementary Information). Figure 1 provides an overview of the experimental setup and conditions, while Figure S4 offers detailed supplementary data that support and elaborate on the findings presented in Figure 1. Viewing these figures together is essential for a comprehensive understanding of the experimental design and the resulting chemical interactions.

(original version, page 3, in the Results and Discussion section)

In order to determine the chemical reactivity of current collectors (Cu, SS, Ni, Al and Al/C), LPSCl sulfide electrolyte and current collectors have been in contact for 24 hours in **coin cell format (CR2032)** as shown in Figure 1 and **Figure S4 (Supplementary Information)**.

11. This is the caption part of Figure (g) in line 120.

Thank you for your insightful comment. Indeed, this caption corresponds to Figure 1g.

12. The front and back of the entire Cu collector should be described, but the table in the figure describes them as pristine and corroded. It seems that the description and the table do not match.

Thanks for your valuable comment. We added some explanation into Figure 7 which you can find below.

(original version, page 16, in the Results and Discussion section)

Figure 7. Lithium plating at a current density of $0.15 \text{ mA}\cdot\text{cm}^{-2}$ of current collectors of Cu, SS and Ni in (a) CC | LPSCl | Li/CC and (b) CC/Li | LPSCl | Li/CC cell formats, and (c) photos of current collectors, LPSCl pellets and lithium metal anode after plating. Note: back side of current collector after LPSCl solid electrolyte contact; one photo of front side shows the surface of current collector after removing the Lithium metal anode; second photo of front side shows together current collector and Lithium metal anode.

13. In line 137, the abbreviation 'didn't' is not appropriate for the written language of a research paper.

Thanks for your valuable comment. We rewrite this part of the text.

(original version, page 6, in the Results and Discussion section)

While for SS and Ni current collectors, the peaks corresponding to the iron (PDF #98-063-1730) and Ni (PDF #96-151-2527) showed no impurities or peak shift: these results confirm the high stability of SS and Ni foil to the chemical reactivity of $\text{Li}_6\text{PS}_5\text{Cl}$ sulfide electrolyte (Figure 2b and 2c).

14. Figures 2, 4, 5, 6 and figures S7, S9 are too small to read. You should divide the figures into larger ones or increase the font size. Published papers will be read by readers in various environments. Considering this, authors should print them out and check that they are readable.

Thank you for your valuable comment. We agree with your suggestions and have revised all figures accordingly. The updated figures are now available in the main body of the manuscript and the Supplementary Information files.

15. It is necessary to present a possible chemical reaction formula and its basis for the byproduct in Figure 2 that is generated by the side reaction between the sulfide solid electrolyte and the current collector.

Thanks for your comment. To address the side reactions between the Li₆PS₅Cl sulfide solid electrolyte and various current collectors (Cu, SS, Ni, Al, Al/C, and Li metal anode), we can consider the following possible chemical reactions and their byproducts:

Copper (Cu): $\text{Li}_6\text{PS}_5\text{Cl} + \text{Cu} \rightarrow \text{Li}_3\text{P} + \text{Li}_2\text{S} + \text{Cu}_2\text{S} + \text{LiCl}$

Stainless Steel (SS): $\text{Li}_6\text{PS}_5\text{Cl} + \text{Fe} \rightarrow \text{Li}_3\text{P} + \text{Li}_2\text{S} + \text{FeS} + \text{LiCl}$

Nickel (Ni): $\text{Li}_6\text{PS}_5\text{Cl} + \text{Ni} \rightarrow \text{Li}_3\text{P} + \text{Li}_2\text{S} + \text{NiS} + \text{LiCl}$

Aluminum (Al): $\text{Li}_6\text{PS}_5\text{Cl} + \text{Al} \rightarrow \text{Li}_3\text{P} + \text{Li}_2\text{S} + \text{Al}_2\text{S}_3 + \text{LiCl}$

Aluminum/Carbon (Al/C): $\text{Li}_6\text{PS}_5\text{Cl} + \text{Al/C} \rightarrow \text{Li}_3\text{P} + \text{Li}_2\text{S} + \text{Al}_2\text{S}_3 + \text{LiCl} + \text{C}$

Lithium Metal Anode (Li): $\text{Li}_6\text{PS}_5\text{Cl} + \text{Li} \rightarrow \text{Li}_3\text{P} + \text{Li}_2\text{S} + \text{LiCl}$

These reactions are based on the decomposition of the Li₆PS₅Cl electrolyte and the formation of various sulfides, phosphides, and chlorides and the specific byproducts depend on the reactivity of the current collector material with the electrolyte. Please find these chemical reaction into the body of text below.

(original version, page 7, in the Results and Discussion section)

Based on the obtained XRD results, the possible formation of chemical reactions between the Li₆PS₅Cl sulfide solid electrolyte and various current collectors (Cu, SS, Ni, Al, Al/C, Cu/Li, and Li metal anode) can be considered. The following chemical reactions and their byproducts are proposed (1-6):

Cu: $\text{Li}_6\text{PS}_5\text{Cl} + \text{Cu} \rightarrow \text{Li}_3\text{P} + \text{Li}_2\text{S} + \text{Cu}_2\text{S} + \text{LiCl};$ (1)

SS: $\text{Li}_6\text{PS}_5\text{Cl} + \text{Fe} \rightarrow \text{Li}_3\text{P} + \text{Li}_2\text{S} + \text{FeS} + \text{LiCl};$ (2)

Ni: $\text{Li}_6\text{PS}_5\text{Cl} + \text{Ni} \rightarrow \text{Li}_3\text{P} + \text{Li}_2\text{S} + \text{NiS} + \text{LiCl};$ (3)

Al: $\text{Li}_6\text{PS}_5\text{Cl} + \text{Al} \rightarrow \text{Li}_3\text{P} + \text{Li}_2\text{S} + \text{Al}_2\text{S}_3 + \text{LiCl};$ (4)

Al/C: $\text{Li}_6\text{PS}_5\text{Cl} + \text{Al/C} \rightarrow \text{Li}_3\text{P} + \text{Li}_2\text{S} + \text{Al}_2\text{S}_3 + \text{LiCl} + \text{C};$ (5)

Li and/or bi-layer Cu/Li: $\text{Li}_6\text{PS}_5\text{Cl} + \text{Li} \rightarrow \text{Li}_3\text{P} + \text{Li}_2\text{S} + \text{LiCl}.$ (6)

These reactions arise from the decomposition of the Li₆PS₅Cl sulfide solid electrolyte, resulting in the formation of various sulfides, phosphides, and chlorides [30,46,47,48]. Therefore, the specific

byproducts depend on the reactivity of the current collector material with the solid electrolyte. However, further detailed investigation is required to confirm these chemical reactions.

16. A detailed explanation is needed as to why the lithium deposition pattern is different in figure 5(c).

Thanks for your comment. In the Current Collector/Li6PS5Cl/Lithium metal anode system, lithium ions from the Li6PS5Cl electrolyte are reduced and deposited onto the copper current collector (for example) during plating, and oxidized back into ions during stripping. Challenges include dendrite formation and uneven stripping, leading to capacity loss. In the Lithium metal anode/Li6PS5Cl/Lithium metal anode system, lithium ions are deposited directly onto the lithium metal anode during plating and oxidized back into ions during stripping. This system also faces issues with dendrite growth and the formation of nanovoids, which can cause micro short circuits and reduce cycling efficiency. The key differences lie in the substrate used for lithium deposition and the associated challenges. We added some points into body of text below.

(original version, page 11, in the Results and Discussion section)

Moreover, it should be noted that in the CC | LPSCl | Li/CC system, lithium ions from the Li6PS5Cl sulfide solid electrolyte are reduced and deposited onto the copper current collector (for example) during the plating process. During stripping, these lithium ions are oxidized back into the solid electrolyte. This system faces challenges such as dendrite formation and uneven stripping, which can lead to capacity loss and short circuits. In contrast, the CC/Li | LPSCl | Li/CC system involves the direct deposition of lithium ions onto the lithium metal anode during plating. During stripping, lithium metal is oxidized back into ions that migrate into the solid electrolyte. This system also encounters issues with dendrite growth and the formation of nanovoids, which can reduce cycling efficiency. The primary differences between these systems lie in the current collector used for lithium deposition and the associated challenges, highlighting the importance of current collector selection in optimizing the performance and safety of solid-state batteries.

17. In lines 39-44 of the introduction, the possible reactions of the solid electrolyte need to be specified and additional references need to be provided.

Thanks for your comment. We added the chemical reaction of the solid electrolytes and current collectors, which you can find below.

(original version, page 7, in the Results and Discussion section)

The following chemical reactions and their byproducts are proposed: Copper (Cu): $\text{Li}_6\text{PS}_5\text{Cl} + \text{Cu} \rightarrow \text{Li}_3\text{P} + \text{Li}_2\text{S} + \text{Cu}_2\text{S} + \text{LiCl}$; Stainless Steel (SS): $\text{Li}_6\text{PS}_5\text{Cl} + \text{Fe} \rightarrow \text{Li}_3\text{P} + \text{Li}_2\text{S} + \text{FeS} + \text{LiCl}$; Nickel (Ni): $\text{Li}_6\text{PS}_5\text{Cl} + \text{Ni} \rightarrow \text{Li}_3\text{P} + \text{Li}_2\text{S} + \text{NiS} + \text{LiCl}$; Aluminum (Al): $\text{Li}_6\text{PS}_5\text{Cl} + \text{Al} \rightarrow \text{Li}_3\text{P} + \text{Li}_2\text{S} + \text{Al}_2\text{S}_3 + \text{LiCl}$; Aluminum/Carbon (Al/C): $\text{Li}_6\text{PS}_5\text{Cl} + \text{Al/C} \rightarrow \text{Li}_3\text{P} + \text{Li}_2\text{S} + \text{Al}_2\text{S}_3 + \text{LiCl} + \text{C}$; Lithium metal anode (Li) and/or bi-layer Cu/Li metal anode: $\text{Li}_6\text{PS}_5\text{Cl} + \text{Li} \rightarrow \text{Li}_3\text{P} + \text{Li}_2\text{S} + \text{LiCl}$.

Thanks for your comment. For these sentences, we added the following references.

(original version, page 2, in the Introduction section)

The reactivity of sulfide electrolytes toward lithium metal and current collectors is still an open question due to the toxic and possible reaction of sulfide components with pure alkaline and transition metals, which can lead to corrosion and result in general capacity degradation of the cell. The manufacture of conventional lithium-ion and solid-state batteries consists mainly of active electrode materials, separators for LIBs, solid electrolytes as separators and electrolytes for SSBs, and current collectors [8,9,10,11,12].

[8] T. Schmaltz, T. Wicke, L. Weymann, P. Voß, C. Neef, A. Thielmann, Solid-State Battery Roadmap 2035+, Fraunhofer ISI: Karlsruhe, Germany, 2022.

[9] Y. Li, M. Lehmann, L. Cheng, T.A. Zawodzinski, J. Nanda, G. Yang, Integrated electro- and chemical characterization of sulfide-based solid-state electrolytes, Mater. Adv. 5 (2024) 9138-9159.

[10] Y. Wu, Z. Zhang, Q. Zhang, Z. Zhang, J. Li, M. Liu, H. Li, L. Chen, F. Wu, Industrialization challenges for sulfide-based all solid state battery, eTransportation 22 (2024) 100371.

[11] D. Ren, L. Lu, R. Hua, G. Zhu, X. Liu, Y. Mao, X. Rui, S. Wang, B. Zhao, H. Cui, M. Yang, H. Shen, C.-Z. Zhao, L. Wang, X. He, S. Liu, Y. Hou, T. Tan, P. Wang, Y. Nitta, M. Ouyang, Challenges and opportunities of practical sulfide-based all-solid-state batteries, *eTransportation* 18 (2023) 100272.

[12] J. Li, Y. Li, Y. Wang, X. Wang, P. Wang, L. Ci, Z. Liu, Preparation, design and interfacial modification of sulfide solid electrolytes for all-solid-state lithium metal batteries, *Energy Storage Materials* 74 (2025) 103962.

18. It is difficult to distinguish between the front side/back side/LPSCI in the ‘after 24h’ pictures in Figure 1. Please indicate them in each picture.

Thanks for your valuable comment. We agree and therefore we added these points into pictures of Figure 1.

(original version, page 5, in the Results and Discussion section)

Figure 1. Photos of current collectors of (a) Cu, (b) SS, (c) Ni, (d) Al, and (e) Al/C before and after contact (in direct contact into coin cell CR2032 **in symmetric system of CC-1 | LPSCI | CC-2**) with LPSCI electrolyte for 24 h and after 2 months: photos of Cu current collector of (f) after contact with LPSCI electrolyte for 24 h, and (g) XRF analysis after 2 months of front and back side of Cu current collector. This experiment simulates direct contact (into coin cell CR2032) of LPSCI pellet and current collectors and the possible formation of side reaction components (impurity phases of Li_2S , LiCl , and/or Li_3PO_4) into coin cell CR2032 which contacted with current collectors and lithium metal anode sources.

19. In figure 1 (f) and (g), it needs to be precise and concise as to whether it is 2 months or 3 months. (Line 102 of the text indicates 2 months, and the caption of Figure 1 (f) says 3 months.)

Thank you for your valuable comment. We agree with your point; it was a typographical error. We actually studied the samples after 2 months. Therefore, we have corrected this point, which you can find below.

(original version, page 5, in the Results and Discussion section)

Figure 1. Photos of current collectors of (a) Cu, (b) SS, (c) Ni, (d) Al, and (e) Al/C before and after contact (in direct contact into coin cell CR2032 **in symmetric system of CC-1 | LPSCI | CC-2**) with LPSCI electrolyte for 24 h and **after 2 months**: photos of Cu current collector of (f) after contact with LPSCI electrolyte for 24 h, and (g) XRF analysis after 2 months of front and back side of Cu current collector. This experiment simulates direct contact (into coin cell CR2032) of LPSCI pellet and current collectors and the possible formation of side reaction components (impurity phases of Li_2S , LiCl , and/or Li_3PO_4) into coin cell CR2032 which contacted with current collectors and lithium metal anode sources.

20. In figure 1 (a) ‘after 24h’, a noticeable surface change is observed, but why are no impurities observed in the XRD data of figure 2 (a)?

Thank you for your valuable comment. Based on the results and data, after one week, we observed a slight shift in the main peaks, indicating a slow reaction between the copper current

collector and the sulfide solid electrolyte. This impact becomes more pronounced in the XRD pattern and on the surface of the copper current collector after one week.

(original version, page 6, in the Results and Discussion section)

In addition, the main peaks of Cu (PDF #98-005-2256) shifted at lower 2θ angles due to the chemical reactions and form side products due to the corrosion of the Cu current collector, indicating a slow reaction between the copper current collector and the sulfide solid electrolyte [32,37]. While this effect becomes more pronounced in the XRD pattern and on the surface of the copper current collector after one week.

21. Why is ‘after 1 week data’ included only in figure 2 (a), and why are there no after 1 week data in (b)-(g)?

Thank you for your valuable comment. We didn’t observe changes in the rest of the current collector compared to the copper and lithium metal anode, likely due to their higher chemical stability or reactivity against sulfide solid electrolyte. Consequently, we believe these data and results may not be relevant or appealing for this paper and its readers, as SS, Ni, Al, and Al/C exhibit low chemical stability. Therefore, we have only included the data and results for Al after 2 months (Figure S5, Supplementary Information) as reference.

22. There is an XRD analysis result in Figure 2, but why are there no pictures of Li and Cu/Li in Figure 1?

Thank you for your comment. The images of indirect contact are very similar to those of direct contact. Therefore, we believe these images may not be valuable for our readers.

23. I would like to mark each picture in Figure 3 with alphabet symbols such as (a), (b)...(i)... and provide a related explanation.

Thank you for your valuable comment. We have added alphabetical symbols to make the content more understandable and readable, and we added some explanations for this Figure 5 which you can find below.

(original version, page 12, in the Results and Discussion section)

Figure 5. Photos of (a) SS, (b) Ni, (c) Cu, (d) Al, (e) Al/C, (f) Li and (g) Cu/Li current collectors after contact (undirect contact into vials) with LPSCl electrolyte after 2 months and after 4 hours heat treatment at 60°C. This experiment simulates the contact of LPSCl pellet and current collectors and the possible formation of side reaction components (impurity phases of Li_2S , LiCl , and/or Li_3PO_4) into vials which are quite toxic and aggressive to current collectors and lithium metal anode sources.

(original version, page 11, in the Results and Discussion section)

This experiment simulates the interaction between the LPSCl pellet and the current collectors and lithium metal anode sources, leading to the potential formation of side reaction products such as Li_3P , Li_2S , LiCl and/or Li_3PO_4 and byproducts depending on the type of current collectors (aforementioned above). These impurity phases are toxic and can aggressively react with the current collectors and lithium metal anodes. Understanding the chemical stability and reactivity of these current collectors when exposed to the LPSCl electrolyte is crucial for assessing their suitability in lithium metal solid-state batteries with sulfide solid electrolyte. The findings provide valuable insights into the degradation mechanisms and help identify the most stable and efficient materials for use in solid-state battery systems. At 60°C, current collectors (SS, Ni, Cu, Al, Al/C) are susceptible to thermal expansion,

increased corrosion, and elevated electrical resistance. In the case of lithium metal anodes (Li, Cu/Li), elevated temperatures can facilitate dendrite growth, electrolyte decomposition, and interfacial instability. Therefore, elevated temperatures can significantly impact chemical decomposition and structural integrity of the current collectors and lithium metal anodes.

24. In all the samples in Figure 4, only the 5h part seems to deviate from the trend. Additional explanation is needed for this data.

Thanks for your valuable comment. We can suppose that after 5 hours (for example) of indirect contact between various current collectors (SS, Cu, Ni, Al, Al/C, Li, Cu/Li) and the Li₆PS₅Cl electrolyte, changes in resistance were observed due to several factors. We can suppose that after 5 hours (for example) of indirect contact between various current collectors (SS, Cu, Al, Ni, Al/C, Li, Cu/Li) and the Li₆PS₅Cl electrolyte, changes in resistance were observed due to several factors. Chemical Reactions: The interaction between the current collectors and the Li₆PS₅Cl electrolyte can lead to chemical reactions that form impurity phases and byproducts can deposit on the surface of the current collectors, altering their electrical properties and increasing resistance. Surface Degradation: The formation of these impurity phases can cause surface degradation of the current collectors which are particularly susceptible to corrosion (for example, Cu, Li), which can significantly impact their conductivity. Solid Electrolyte Interphase (SEI) Formation: The interaction with the sulfide solid electrolyte can lead to the formation of a solid electrolyte interphase (SEI) layer. This layer consists of various byproducts from the electrolyte and the current collectors, which can increase resistance by creating a barrier to electron flow. Material Stability: Different materials exhibit varying degrees of stability when exposed to or in contact with the sulfide solid electrolyte. For example, SS, Ni, Al, and Al/C tend to maintain higher stability compared to Cu and Li, which show more pronounced resistance changes due to their higher reactivity. Therefore, these factors contribute to the observed changes in resistance indirect or indirect contact with the Li₆PS₅Cl electrolyte, highlighting the importance of selecting chemically stable materials for current collectors in solid-state battery systems with sulfide solid electrolytes. Therefore, we added some explanation into body of the text below.

(original version, page 13, in the Results and Discussion section)

After several hours of indirect or undirect contact between various current collectors (SS, Cu, Ni, Al, Al/C, Li, and Cu/Li) and the Li₆PS₅Cl sulfide solid electrolyte, changes in resistance were observed due to several factors such as chemical reactions between the current collectors and the LPSCl electrolyte led to the formation of impurity phases, which deposited on the surface and increased resistance. Surface degradation, particularly in Cu, Li and Cu/Li, further impacted conductivity. Additionally, the formation of a solid electrolyte interphase (SEI) layer created a barrier to electron flow, contributing to increased resistance. Materials like SS, Ni, Al and Al/C maintained higher stability compared to Cu, Li and Cu/Li which showed more pronounced resistance changes due to their higher chemical reactivity. These observations highlight the importance of selecting chemically stable materials for current collectors in sulfide solid-state battery systems.

25. The number of cycles (10 times) in Figure 6 is too short. This study is about the long-term chemical stability of the current collector and solid electrolyte for more than 3 months, so it is necessary to present long-life cycle data accordingly.

Thank you for your valuable comment. We concur with the observations regarding the long-term cycling life of half-cells with various current collectors. After 10 cycles, some degradation of the cells was observed as shown in Figure 8. Therefore, we believe these preliminary data are sufficient to demonstrate the degradation and stability of the current collectors.